# A machine learning approach for estimating forage maize yield and quality in NW Spain

Silverio García-Cortés [1]*, Agustín Menéndez-Díaz [2], María José Bande-Castro [3‡],
Alfonso Carballal-Samalea [4‡], Adela Martínez-Fernández [4‡],
Jose Alberto Oliveira-Prendes [5]

**1** Cartographic Engineering Area, University of Oviedo, Asturias, Spain, **2** Construction and Manufacturing Engineering Dept, University of Oviedo, Asturias, Spain, **3** Grassland and Crop Dept, Agricultural Research Center of Mabegondo, Galician Agency of Food Quality (Agacal), Galicia, Spain, **4** Regional Service for Agri-food Research and Development, Asturias, Spain, **5** Grassland ad Forage Research Program. Plant Production Area, University of Oviedo, Asturias, Spain

☯ These authors contributed equally to this work.
‡ MJB-C, AC-S and AM-F also contributed equally to this work.
* sgcortes@uniovi.es

## Abstract

Crop models simulate crop growth and development according to different climatic, soil and crop management conditions. The CSM-CERES-Maize model (DSSAT) was adapted to simulate forage maize yields by calibrating the genetic parameters of six cultivars: SE1–200, SE2–300 and SE3–400 in three sites and three years in Asturias, and XU1–220, XU2–300 and XU3–400 in four sites and three years in Galicia. Calibration using the CSM-CERES-Maize model, together with the use of historical meteorological data (2000–2022) from the study sites, enabled simulation of forage maize yield (whole plant dry matter yield) and quality (whole plant net energy for lactation yield and whole plant crude protein yield) for six cultivars during the 23-year period. LightGBM models (a machine learning technique) were used with the simulated forage maize yield, quality data, historical weather, soil, and management data to capture non-linear relationships in the data and to identify the most influential variables for crop yield and quality predictions. The results of the model evaluation yielded an accuracy of 94.7%, ($R^2$ score = 0.86) for forage maize yield, an accuracy of 94.0% ($R^2$ score = 0.84) for the net energy for lactation yield and an accuracy of 93.0% ($R^2$ score = 0.85) for the crude protein yield. Variable importance plots revealed Growing Season and Radiation from sowing to harvest to be the top two most influential predictor variables. In Asturias and Galicia, the cultivars with the longest cycle (cultivars cycle 400) are those with the highest values for the variables studied in the 23 years of historical meteorological data (average of three sites in Asturias and four sites in Galicia with three sowing dates in each site). The models will be available to make predictions for forage maize yield and quality by non-specialist users, using the geographical location of the crop field, cultivar type, sowing and harvest date and probable values of weather variables during the growing season as input data.

**Data availability statement:** Data & Code relevant to this paper is available from: https://doi.org/10.5281/zenodo.15470090.

**Funding:** This research was supported by a mobility research grant awarded to JAO (TAD/CRP PO 500109615) from the OECD Co-operative Research Programme. The funders had no role in study design, data collection and analysis, decision to publish, or preparation of the manuscript.

**Competing interests:** The authors have declared that no competing interests exist.

## 1. Introduction

Production of forage maize in Spain is concentrated in the northern part of the country. The main production regions are Galicia, Asturias and Cantabria, covering areas of 73836 ha, 7033 ha and 4610 ha respectively [1]. Together, these three regions account for 89% of the area planted with forage maize in Spain.

In collaboration with seed-producing companies, an evaluation of commercial hybrid maize cultivars for silage was initiated in 1996 in Asturias by the Regional Service for Agri-Food Research and Development (SERIDA) and in 1999 in Galicia by the Agricultural Research Centre of Mabegondo (CIAM) [2,3] and has been continued to the present. High levels of interannual variability in the results can occur owing to differences in climatology (temperature, precipitation, etc.). It is therefore important to have data available for more than one year to enable agronomic characterization of cultivars.

The sensitivity of maize grain yield to elevated temperatures (alone or associated with water or nutrient stress) is much higher during the critical flowering period, which determines the number of grains per plant or during grain filling, which influences grain weight and quality [4].

Process-based (functional) models are excellent tools for studies quantifying the effects of management, genetics, soil and climate on crop yields and phenology. The CSM-CERES-Maize model [5–7] within the same DSSAT package has been widely used to support decision-making regarding the management of irrigation and fertilization, as well as the choice of maize cultivars. CERES-Maize continues to be the most widely used maize model globally and remains the basis of other maize models, including APSIM [8] and the CSM-IXIM [9].

In forage maize, [10,11] initially adapted the CSM-CERES-Maize model (CSM = Cropping System Model) in the software provided by the Decision Support System for Agrotechnology Transfer (DSSAT) [12] for forage maize, in order to simulate the growth and development of three forage maize cultivars (SE1–200, SE2–300, and SE3–400) in three sites (Barcia, Villaviciosa and Grado) in Asturias.

One of the major problems in the selection of genotypes (cultivars) with high productivity in different environments (locations, years) is the genotype x environment interaction (GEI). Multi-environment trials (MET) often use the AMMI (additive main-effects and multiplicative interaction) model which is popular for analyzing MET data with fixed effect [13]. This model is a statistical tool for identifying GEI patterns and allows grouping genotypes according to response characteristics (identification of stable genotypes) and detecting trends between environments [14]. Moreover, DSSAT-based seasonal analysis was conducted to examine the interannual variation in forage maize productivity in combination with meteorological data available for 23 years (2000–2022) in the three sites in Asturias. Similar work was later performed with data from three cultivars (XU1–200, XU2–300, and XU3–400) in four sites (Ribadeo, Ordes, Deza and Sarria) in Galicia [15].

The objective of the present study was to obtain a predictive model using a forage maize dataset (field, weather, soil and management data) and a machine learning technique to help optimize agronomic practices and harvest decisions for forage maize farmers and policymakers in Northwestern Spain.

The workflow carried out in this case from old and new data is summarized in the following chart (Fig 1).

The structure of this article is organized as follows: the Materials and methods section details the field data collected across seven localities in the northwest of Spain, along with the adaptation of the CSM-CERES Maize model to simulate forage maize yield and quality by calibrating the genetic parameters of six cultivars. Additionally, the methodology for generating synthetic data to analyze interannual variations in production over a 23-year period is described. Subsequently, the Machine Learning Processing subsection describes the LightGBM model and its application in developing a predictive model for forage maize yield based on the data presented in the previous section. The Results and discussion section presents the model validation metrics obtained using the reserved dataset for this purpose. Finally, the study concludes with a summary of key findings and insights in the final section.

## 2. Materials and methods

### 2.1 Experimental sites and minimum data sets in Asturias and Galicia

Maize cultivars (SE1–200, FAO-200; SE2–300, FAO-300; SE3–400, FAO-400) were evaluated in field trials conducted in three sites in Asturias: Barcia (43.5402, −6.4954 and 25 masl), Villaviciosa (43.4722, −5.4361 and 10 masl) and Grado (43.3764, −6.0625 and 50 masl). The cultivars are given fictitious names here due to the strict confidentiality regarding the name of the hybrids imposed by the seed companies.

The soil in the Barcia site, located on the western coast of Asturias, is characterized as a loam soil (order Inceptisol, suborder Udepts, great group Dystrudepts). The soil in Villaviciosa, in the central coastal zone, has a clay-loam texture (order Entisol, suborder Fluvents, great group Udifluvents). The soil in Grado, situated in an interior valley in central Asturias, has a sandy clay loam texture (order Inceptisol, suborder Udepts, great group Dystrudepts) [16].

All the study sites belong to the temperate ocean climate zone (type Cfb), according to the Köppen-Geiger climate classification [17]. The temperature in the coldest month is lower than 18 ºC but higher than −3 ºC. The mean temperature

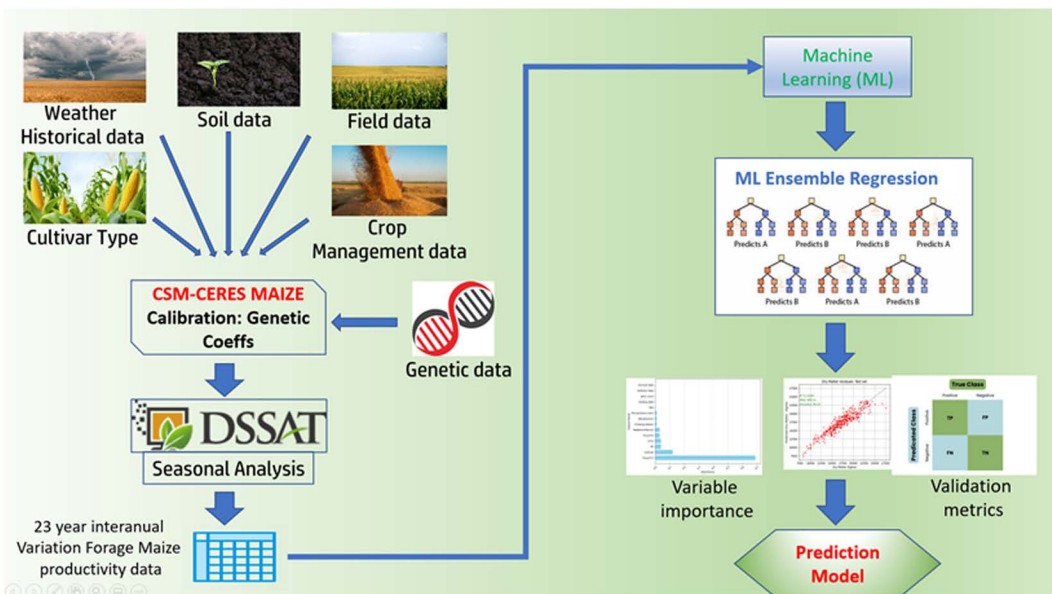

**Fig 1. General workflow.** Left vertical part represents the field data sources and synthetic data creation using CSM-CERES and DSSAT. Right part shows the Machine learning modelling phase of the previous data. The outputs of the process are a forage maize production prediction model, associated validation metrics and input variable relative importance for those predictions.

is lower than 22 °C in all months and higher than 10 °C in at least four months of the year. The precipitation does not vary significantly between seasons [18].

Maize cultivars (XU1–200, FAO-200; XU2–300, FAO-300; XU3–400, FAO-400) were evaluated in experimental field trials at four locations in Galicia: Ribadeo (43.5458, −7.0816 and 43 masl), Ordes (43.0432, −8.4458 and 300 masl), Deza (43.6995, −8.3192 and 400 masl) and Sarria (42.8194, −7.3758 and 520 masl). The soils included in the trials in Ribadeo (located in A Mariña Oriental, north-east of Lugo), Ordes (centre of the province of A Coruña) and Deza (north of Pontevedra) have a sandy loam texture. The Ribadeo soil is developed on slate and those of Ordes and Deza on schist. The soil in the Sarria (south of Lugo) trials has a sandy-clay loam texture and is classified as slate soil.

The Ribadeo experimental site, according to the Köppen-Geiger climate classification, belongs to the humid temperate climate with warm summer (type Cfb). The average temperature of the coldest month is below 18 °C and above −3 °C.

The average temperature in the hottest month does not reach 22 °C, but it is higher than 10 °C for four or more months of the year. Rainfall is spread throughout the year, and there is no dry season. On the other hand, the other three experimental sites belong to the temperate rainy climate with dry and warm summer (Csb). The mean temperature of the coldest month is below 18 °C and above −3 °C. The mean temperature of the warmest month does not reach 22 °C and exceeds 10 °C in four or more months of the year. Precipitation exceeds evaporation. Rainfall decreases considerably in summer, coinciding with high temperatures.

A historical series of meteorological data for the experimental sites, covering a period of 23 years, was obtained at the weather stations closest to the experimental sites and provided by the State Meteorological Agency [19] for Asturias and by the regional MeteoGalicia [20] for Galicia.

## 2.2 Cultivar characteristics

In Table 1, the values of the sowing date (the day of the year = doy or Julian date), anthesis date (doy), harvest date (doy) and the periods between sowing and anthesis date and between sowing and harvest date (Growing season) can be seen.

## 2.3 Adaptation of the CSM-CERES-Maize model

CSM-CERES-Maize requires six parameters, known as "genetic coefficients", to characterize different cultivars. Each genetic coefficient has a direct influence on a specific crop model variable (Table 2).

The estimated genetic coefficients for three cultivars (SE1–200, SE2–300, SE3–400) for three locations in Asturias and for three cultivars (XU1–200, XU2–300, XU3–400) for four locations in Galicia for three years are indicated in Table 3.

At present, CSM-CERES-Maize simulates seven phenological states: germination, emergence, end of the juvenile phase (in maize, leaf 6 and leaf 7 are juvenile-to-adult transition leaves and leaf 8 is normally an adult leaf according to [21], flower initiation (anthesis date), 75% of the plants with visible stigmas (female flowering), initiation of grain filling and physiological maturity. These stages do not give sufficient details to produce forage maize destined for silage, as

**Table 1. Mean values and Standard deviations in brackets of hybrid forage maize cultivars (SE1-200, SE2-300 and SE3-400 in Asturias and XU1-200, XU2-300 and XU3-400 in Galicia).**

| Cultivar | Sowing date (doy) | Anthesis date (doy) | Harvest date (doy) | Period from Sowing-Anthesis (days) | Growing season (days) |
|---|---|---|---|---|---|
| SE1–200 | 143 (11.1) | 212 (7.5) | 259 (11.5) | 70 (7.7) | 116 (6.8) |
| SE2–300 | 143 (11.1) | 225 (8.8) | 268 (10.8) | 82 (5.2) | 125 (4.3) |
| SE3–400 | 143 (11.1) | 227 (5.5) | 273 (10.1) | 84 (8.2) | 130 (6.1) |
| XU1–200 | 139 (6.2) | 208 (3.6) | 254 (3.9) | 69 (7.6) | 116 (8.6) |
| XU2–300 | 139 (6.2) | 217 (6.3) | 265 (7.5) | 78 (8.5) | 127 (10.7) |
| XU3–400 | 139 (6.2) | 220 (7.0) | 268 (8.3) | 82 (10.1) | 129 (11.3) |

**Table 2. Genetic coefficients that characterize each cultivar type in the CSM-CERES-Maize model.**

| Parameter | Definition | Unit | Variables directly influenced by the parameter |
|---|---|---|---|
| P1 | Thermal time from seedling emergence to the end of juvenile phase (expressed in degree days above a base temperature of 8 ºC) during which the plant is not responsive to changes in photoperiod | ºC day | Anthesis date |
| P2 | Extent to which development (expressed as days) is delayed for each hour increase in photoperiod above the longest photoperiod at which development proceeds at a maximum rate (which is 12.5 h) | days | Anthesis date |
| P5 | Thermal time from silking to physiological maturity (expressed in degree days above a base temperature of 8 ºC) | ºC day | Physiological maturity date |
| G2 | Maximum possible number of grains per plant | grains | Number of grains per plant |
| G3 | Grain filling rate during the linear grain filling stage and under optimum conditions | mg/day | Grain dry weight |
| PHINT | Phyllochron interval; the interval in thermal time (degree days) between successive leaf tip appearances | ºC day | Final leaf number |

**Table 3. Estimated genetic coefficients in three cultivars (SE1-200, SE2-300, SE3-400) in 3 years and 3 locations in Asturias and three cultivars (XU1-200, XU2-300, XU3-400) obtained with experimental data from 3 years and 4 locations in Galicia.**

| Cultivar | P1 (ºC day) | P2 (days) | P5 (ºC day) | G2 (grains) | G3 (mg/day) | PHINT (ºC day) |
|---|---|---|---|---|---|---|
| SE1–200 | 145 | 0.3 | 705 | 650 | 6 | 40 |
| SE2–300 | 215 | 0.3 | 650 | 650 | 7 | 40 |
| SE3–400 | 230 | 0.3 | 660 | 650 | 7 | 40 |
| XU1–200 | 115 | 0.3 | 590 | 650 | 7 | 40 |
| XU2–300 | 160 | 0.3 | 590 | 650 | 8 | 40 |
| XU3–400 | 175 | 0.3 | 580 | 650 | 8 | 40 |

harvesting is determined in the field because of the position of the milk line in the grain. This variable (not simulated by the CSM-CERES-Maize model) is commonly used as an indicator of the optimal moisture content for harvesting forage maize for silage [22,23]. To overcome the model limitations, it was assumed that at the time of harvesting the forage maize that the milk line in the grain is halfway between the crown and the tip, i.e., 13 days before physiological maturity, as indicated by [24].

The CSM-CERES-Maize model was calibrated with data obtained in the 2012-2013-2014 and 2014-2015-2016 field trials at the three and four evaluation locations in Asturias and Galicia respectively [25].

## 2.4  Simulation of the interannual variation in forage maize production

With the aim of examining the interannual variation in the anthesis date, dry matter production in the whole plant and nitrogen production in the whole plant, the model was executed with historical meteorological data for a period of 23 years (2000–2022), used to simulate the production and phenology of the cultivars in each of the study locations in Asturias and Galicia [25].

In Spain and France, net energy in feed is expressed in a barley feed unit (one kg of standard barley contains one Unité Fourragère, one UF). Net energy represents the energy used by the animal's body for maintenance, growth and production. The net energy for lactation (NEL) in dairy cattle is also given according to the French standards and expressed as UFL (UF Laitières = milk forage unit = NEL, with 1 UFL = 7.1 MJ/kg DM = 1.7 Mcal NEL) according to [26]. UFL values are obtained from laboratory determination of in vitro digestibility of organic matter and organic matter in the feeds, therefore they have a high correlation with feed digestibility.

As the current CSM-CERES-Maize model does not enable calculation of the net energy of the maize forage (UFL/kg DM) and whole plant net energy for lactation yield (UFL/ha), the dry matter production of the whole plant was converted into energy production by considering the ear harvest index (HIPD = ear dry matter production/total biomass production), as demonstrated by [27].

Values of 0.61 UFL/kg DM and 1.08 UFL/kg DM were used for the energy contents (energy value) of the foliage (stems + leaves + husks) and ears respectively. The following equations were used:

$$\frac{UFL}{kg\ DM} = HIPD \times 1.08 \left[\frac{UFL}{kg\ DM_{ear}}\right] + (1 - HIPD) \times 0.61 \left[\frac{UFL}{kg\ DM_{foliage}}\right] \tag{1}$$

$$\frac{UFL}{ha} = \frac{UFL}{kg\ DM} \times \frac{kg\ DM}{ha} \tag{2}$$

The CSM-CERES-Maize model enables calculation of the kg N/ha but not of the kg CP/ha. Almost all the N in plants is present as amino acids in proteins and the average N content of the proteins (16%) is therefore 100/16 = 6.25 [28]. The crude protein yield (kg CP/ha) was calculated as the kg N/ha x 6.25.

The description of the variables included in the Asturian and Galician forage maize dataset used in the machine learning technique are presented in Table 4.

**Table 4. Description of the variables included in the Asturian and Galician forage maize dataset generated with the CSM-CERES-Maize model.**

| Variable | Description |
|---|---|
| Experimental site (Elevation, m) | Asturias: Barcia (25 m), Villaviciosa (10 m), Grado (50 m)<br>Galicia: Ribadeo (43 m), Ordes (300 m), Deza (400 m), Sarria (520 m) |
| Period | 2000-2022 |
| Cultivar | Asturias: SE1–200, SE2–300, SE3–400<br>Galicia: XU1–200, XU2–300, XU3–400 |
| **Weather data** | |
| Tmax (ºC) | Average daily maximum air temperature from sowing to harvest |
| Tmin (ºC) | Average daily minimum air temperature from sowing to harvest |
| Precipitation (mm) | Total precipitation from sowing to harvest |
| Solar radiation (MJ/m$^2$) | Total solar radiation from sowing to harvest |
| **Soil data** | |
| pH | -$\log_{10}$([H$^+$]) |
| C (%) | Soil organic carbon (%) |
| WHC (mm) | Plant available water capacity (mm) |
| **Important dates** | |
| Sowing dates (Julian date) | Asturias: 133 = 12/05, 136 = 15/05, 151 = 30/05<br>Galicia: 136 = 15/05, 151 = 30/05, 167 = 15/06 |
| Anthesis dates (Julian date) | |
| Harvest dates (Julian date) | Asturias: 263 = 19/09, 267 = 23/09, 287 = 13/10<br>Galicia: 250 = 06/09, 262 = 18/09, 267 = 23/09 |
| Growing season (days) | Number of days from sowing to harvest |
| **Response variables** | |
| kg DM/ha | Whole plant dry matter yield at harvest |
| UFL/ha | Whole plant net energy for lactation yield at harvest |
| kg CP/ha | Whole plant crude protein yield at harvest |

## 2.5 Genotype × environment × management interaction

A three-way ANOVA was conducted to determine the effects of Cultivar, Site and Sowing date on dry matter yield. The three independent variables or factors (Cultivar, Site and Sowing date) were considered fixed factors.

To find out information about the three-way Site x Cultivar x Sowing date interaction (e.g., whether the three-way interaction effect is statistically significant), we need to consult the "Site x Cultivar x Sowing date" row in the Table 5.

There was no statistically significant three-way interaction between Cultivar, Site and Sowing date, $F_{(24, 1386)} = 1.287$, $p \geq 0.05$, but all the two-way interactions were significant.

There was a statistically significant two-way interaction effect between Cultivar and Site, on dry matter yield of the maize cultivars, $F_{(12, 1386)} = 5.288$, $p < 0.001$. This indicates that cultivars were affected differently by the Sites. There was also a statistically significant two-way interaction effect between Site and Sowing date, $F_{(12, 1386)} = 5.775$, $p < 0.001$ and Cultivar and Sowing date, $F_{(4, 1386)} = 4.616$, $p < 0.001$ on dry matter yield.

We may follow up and interpret the two-way interactions but not the main effects due to the statistical significance of the two-way interactions. Usually when we have a significant two-way interaction (e.g., Site x Cultivar), it is the effect of this interaction that is of interest, and the main effects (e.g., Site and Cultivar) are less of interest, because, in this case, we know that the effect of Cultivar changes across levels of Site. There was a statistically significant simple main effect of Site, $F_{(6, 1386)} = 201.6$, $p < 0.001$, Cultivar, $F_{(2, 1386)} = 447.9$, $p < 0.001$, and Sowing date, $F_{(2, 1386)} = 617.9$, $p < 0.001$, on Dry matter yield.

The graphical analysis (not presented here) showed non-crossover interactions [29] indicating that the difference in performance (kg DM/ha) of the Cultivars is not similar across the other factors (Sites or Sowing dates) but it does not change the order (ranking) of the one that produces more and the one that produces less according to the Sites or Sowing dates.

## 2.6 Machine learning processing

The goal is to utilize machine learning techniques to develop a model tailored to this geographic region that can provide production predictions using simple variables associated with the given locations, weather forecasts, and crop management practices. This approach would enable producers or agricultural managers to simulate production scenarios in advance without requiring in-depth knowledge of complex agronomic parameters, such as those governing functional models like CSM-CERES-Maize or the genetic calibration parameters needed for forage maize adaptations. A machine learning technique, such as the ones used in this study, can construct a predictive model for maize production based on these input variables. This approach eliminates the need to explicitly model the physical functional relationships between

Table 5. Summary of Three-Way Analysis of Variance for Site, Cultivar and Sowing date factors on dry matter yield (kg DM/ha).

| Source of variation | df | Mean Square | F |
|---|---|---|---|
| Site | 6 | 737004122.3 | 201.6*** |
| Cultivar | 2 | 1637578561.7 | 447.9*** |
| Sowing date | 2 | 2259102449.8 | 617.9*** |
| Site x Cultivar | 12 | 19332573.5 | 5.3*** |
| Site x Sowing date | 12 | 21113139.5 | 5.8*** |
| Cultivar x Sowing date | 4 | 16876285.6 | 4.6*** |
| Site x Cultivar x Sowing date | 24 | 4703423.8 | 1.3NS |
| Error | 1386 | 3655797.3 | |
| Total | 1449 | | |

NS = no significant (p ≥ 0.05), *** = significant at p < 0.001.

df = degrees of freedom, F = F-test.

variables and can establish fundamentally complex and non-linear relationships between the input and output variables with suitable precision. The field data were observed during the years 2012–2013–2014 in Asturias and 2014–2015–2016 in Galicia. Out of the total 23 years of data, 6 years correspond to field observations and the rest are simulated, meaning that approximately 25% of the data are field based. This proportion of field data versus simulated data is expected to be maintained in both the training and test sets.

**2.6.1 Programming language and used packages.** Python has been used as the programming language within *Jupyter Notebook* environment. The Python packages used include: *numpy, pandas*, for basic programming and *sklearn, lightgbm, xgboost, optuna, shap* for machine learning processes. Other auxiliary packages*: joblib, matplotlib, streamlit,* and *plotly*, were used for graphics and file outputs. Data and code links are supplied in the *Supporting Information* section in S1 File.

**2.6.2 Exploratory Data Analysis (EDA).** EDA is the first step in almost every Machine learning study and helps to understand the structure, quality, and characteristics of the dataset, ensuring the machine learning models are built on reliable and meaningful data.

Summary statistics of the variables used in the machine learning model are shown in Table 6. These variables were obtained after calibrating and evaluating the CSM-CERES-Maize (DSSAT) model to simulate forage maize production in Asturias and Galicia. Adaptation of the model, together with the use of historical meteorological data (2000–2022) from the study sites, enabled simulation for the whole plant dry matter yield, the whole plant milk forage unit yield and the whole plant crude protein yield of the six cultivars during the 23-year period.

Table 6. Mean values of the variables included in the Asturian and Galician forage maize dataset and the associated standard deviations (SD) for three forage maize cultivars in three locations: Barcia, Villaviciosa and Grado and three years 2012, 2013 and 2014 in Asturias and three forage maize cultivars in four locations: Ribadeo, Ordes, Deza and Sarria and three years 2014, 2015 and 2016 in Galicia.

| Variable | Mean | SD |
|---|---|---|
| Elevation (m) | 193.0 | 194.9 |
| **Weather data** | | |
| Tmax (ºC) | 23.7 | 2.0 |
| Tmin (ºC) | 13.8 | 2.1 |
| Precipitation (mm) | 172.0 | 73.0 |
| Solar radiation (MJ/m$^2$) | 2193.0 | 382.2 |
| **Soil data** | | |
| pH | 6.1 | 0.5 |
| C (%) | 2.8 | 1.2 |
| WHC (mm) | 97.0 | 13.7 |
| **Important dates** | | |
| Sowing dates (Julian date) | 151.0 | 13.2 |
| Anthesis dates (Julian date) | 224.0 | 12.9 |
| Harvest dates (Julian date) | 265.0 | 10.8 |
| Growing season (days) | 114.0 | 17.2 |
| **Response variable** | | |
| kg DM/ha | 17527.0 | 3520.7 |
| UFL/ha | 15373.0 | 3215.8 |
| kg CP/ha | 1110.0 | 243.0 |

The mean values obtained for these variables are within the values obtained by the evaluation network of forage maize varieties in Asturias and Galicia [30,31].

Good quality long-term station data plays a significant role in characterizing the climatic conditions and in assessing their suitability for agricultural production [32]. Recent studies highlight the importance of using gridded meteorological datasets as reliable alternatives to directly measured data, especially in areas with sparse weather station coverage [33]. Examples of available meteorological databases include the following: NASA Power from the NASA Langley Research Center POWER (Prediction Of Worldwide Energy Resources) Project (https://power.larc.nasa.gov/) provides solar and meteorological data sets from NASA research for support of renewable energy, building energy efficiency and agricultural needs and CHIRPS (Climate Hazards Group InfraRed Precipitation with Station data) from the Climate Hazards Center, University of California, Santa Barbara for precipitation data (https://www.chc.ucsb.edu/data/chirps) [34].

In Fig 2, in this case we performed a clustermap analysis to understand the linear correlations between the variables and the hierarchies of the different groups of them with respect to the target production variables. The color and numbers in the cells of the chart inform about the linear correlations between teach input variable (rows) and the target one (columns). Also, the lines in the margin give a sense of variable groups by similar content of information with respect to the prediction target values. Radiation and Growing season showed the bigger positive correlation meanwhile Sowing date present the bigger in negative terms. The chart in Fig 2 shows that there are roughly two big groups of input variables which do not clearly agree with physical meanings like soil and meteorological group of vars. In addition, linear correlations between variables are high only for few vars like Radiation and Growing season duration which can be modelling the same type of information.

Meteorological variables for Asturias locations and for Galicia ones can be seen in Fig 3. For Asturias, the average values of meteorological variables over 23 years in the forage maize crop cycle (sowing-harvesting) for Tmax (22.9 °C, SD = 1.41), Tmin (15.3 °C, SD = 1,25), Total solar radiation (2296 MJ/m$^2$, SD = 320.7) and Total precipitation (208 mm, SD = 76.8 mm) were similar to those obtained in Galicia during the maize crop cycle for Tmax (24.3 °C, SD = 2.22) and Tmin (12.7 °C, SD = 1.97) and slightly better (higher values) in terms of Total Solar radiation (2115 MJ/m$^2$, SD = 405.7) and Total precipitation (145 mm, SD = 57.0). The maximum and minimum temperature values obtained in the 23 years of historical meteorological data in Asturias and Galicia are within the ranges considered suitable for maize cultivation [35]. Maximum temperatures (Tmax) registered during the 23-year period in the seven sites of experimentation were always below 30 °C. Brief or prolonged episodes (more than 5 days) of high temperatures stress (>35 °C), especially at the flowering stage (longer anthesis-silk interval), lead to reduced yields [[36–38]].

In Asturias (Fig 4), the highest value for the whole plant dry matter yield (22887 kg DM/ha) was obtained in 2018 with cultivar 400 and average daily maximum temperatures of 21.9–25.2 °C, average daily minimum temperatures of 14.5–17.4 °C, total solar radiation of 2148–2772 MJ/m$^2$ and total rainfall of 195–412 mm. In Galicia (Fig 4), the highest value for the whole plant dry matter yield (19637 kg DM/ha) was obtained in 2000 for cultivar 400, with average daily maximum temperatures of 22.3–27.0 °C, average daily minimum temperatures of10.7–15.9 °C, total solar radiation of 1960–2987 MJ/m$^2$ and total rainfall of 87–236 mm.

**2.6.3 Regressor Machine Learning algorithms tested in the study. 2.6.3.1 Random Forest Regressor:** Random Forest Regressor (RFR) [39] is a supervised ensemble learning algorithm used for regression and classification tasks. It builds multiple decision trees using random subsets of data and features and averages their predictions to improve accuracy and reduce overfitting. Each tree partitions the feature space to minimize a loss function, typically the mean squared error (MSE) in regression. RFR is robust to outliers and requires tuning of hyperparameters such as the number of trees and tree depth. In this study, hyperparameter optimization was performed using Optuna. Feature importance was also computed to evaluate the contribution of each predictor. These methods have already been used for maize model prediction [40].

One of the drawbacks of RFR is the requirement for tuning hyperparameters like the number of trees and the maximum depth of each tree to achieve optimal performance. In this case we performed a Randomized SearchCV [41] by randomly

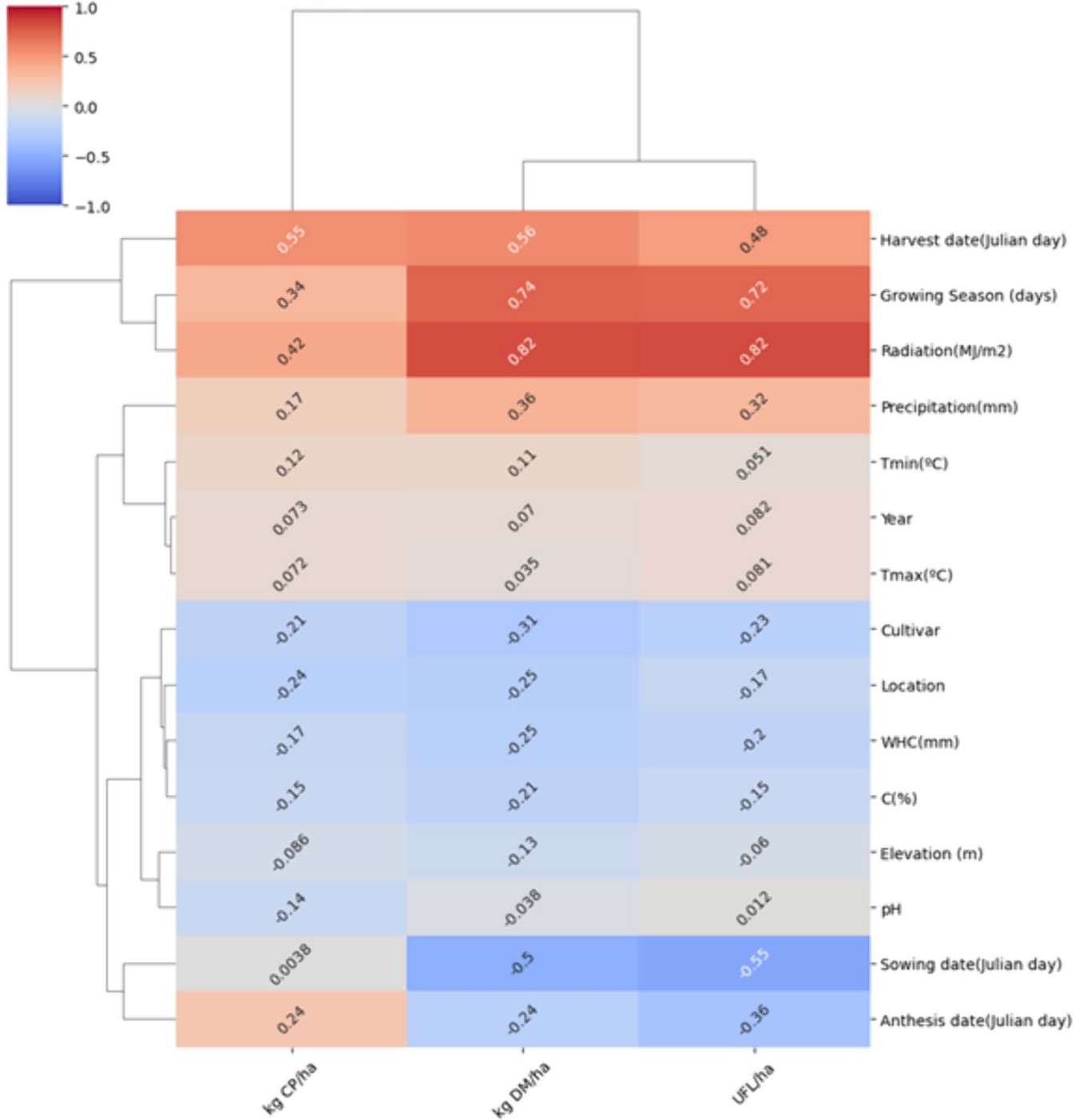

**Fig 2. Dendrogram chart, grouping similar input variables and linear correlation values with respect to production vars.** Input variables by rows and target variables by columns.

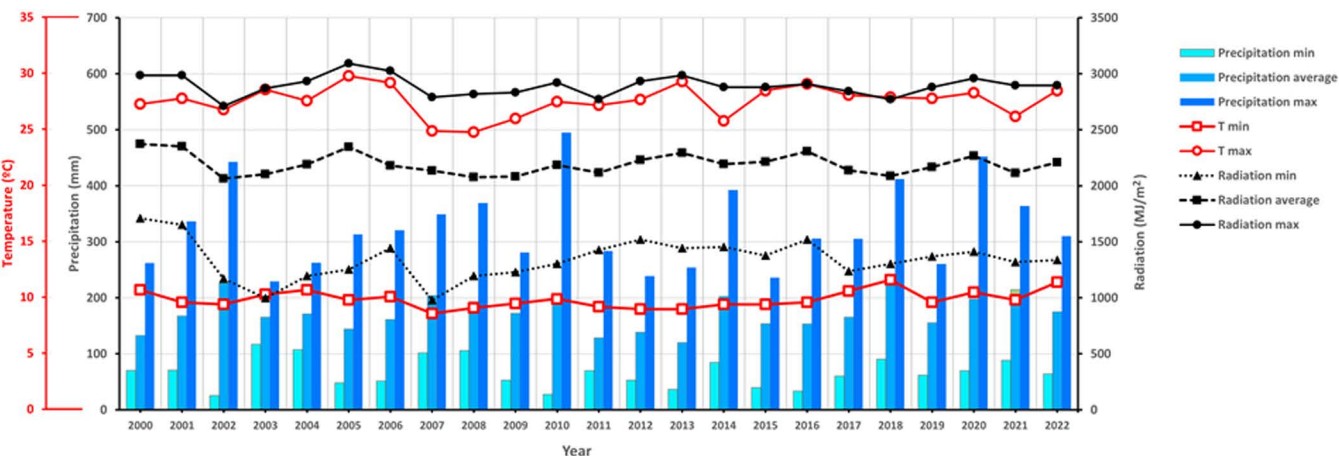

**Fig 3. Historical weather data (Tmax, Tmin, Solar radiation and Precipitation) for 2000-2022 period in Asturias and Galicia study locations.**

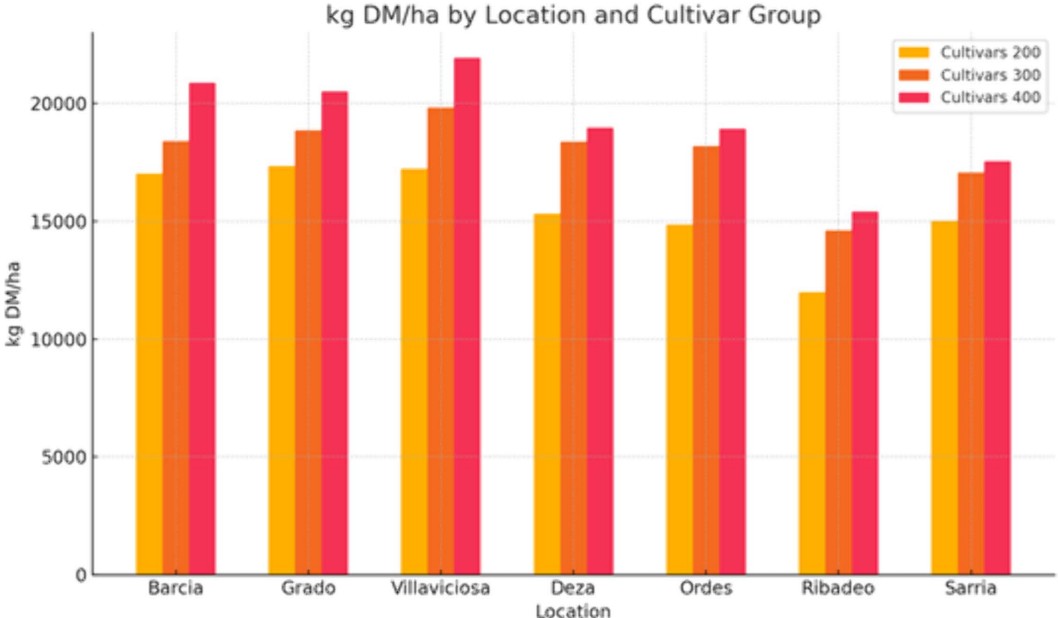

**Fig 4. Comparison of whole plant dry matter yield (kg DM/ha) simulated using 23 years of meteorological data in Asturias (Barcia, Grado, Villaviciosa) and Galicia locations for cultivars 200, 300 and 400.**

sampling hyperparameter values from a specified search distribution for each hyperparameter. This process evaluates the performance of the model with these random combinations and iterates for a defined number of times [42]. Finally, the importance feature is obtained to assess the relative amount of information extracted from each feature during model training.

**2.6.3.2 LightGBM:** LightGBM (Light Gradient Boosting Machine) [43] is a gradient boosting framework based on decision trees that is optimized for speed and efficiency. It grows trees leaf-wise (as opposed to level-wise) with depth constraints, allowing better accuracy and faster training on large datasets. LightGBM handles categorical variables natively

and supports parallel and GPU learning. It is particularly effective for high-dimensional data. In this study, hyperparameter tuning was performed using Optuna, and feature importance was computed to assess predictor contributions.

**2.6.3.3 XGBoost:** XGBoost (Extreme Gradient Boosting) [44] is a scalable, regularized boosting algorithm that builds additive regression trees using gradient descent to minimize a specified loss function. It includes built-in regularization to prevent overfitting and supports parallel computation, missing value handling, and customizable loss functions. XGBoost is known for its strong performance in structured data problems. In this work, we optimized hyperparameters using Optuna and evaluated global feature importance to interpret the model's behavior.

**2.6.3.4 Adaboost:** AdaBoost (Adaptive Boosting) [45] is an ensemble learning method that combines multiple weak learners, typically shallow decision trees into a stronger predictive model. It builds models sequentially, with each new learner focusing on the samples most misclassified by previous ones. AdaBoost adjusts the weights of training examples to emphasize difficult cases. While simple and effective for certain problems, it is more sensitive to noisy data and outliers. In this study, we included AdaBoost for comparison, although it showed lower predictive performance compared to the other methods.

**2.6.4 Model performance.** The following metrics were used to evaluate the predictive quality of the models: $R^2$ coefficient, RMSE (Root Mean Squared Error), MAE (Mean Absolute Error), MAPE (Mean Absolute Percentage Error) and Accuracy (for regression). The metrics are defined in Table 7.

**2.6.5 Optuna: Hyperparameter optimization.** Optuna [46] is an automatic hyperparameter optimization framework that uses intelligent sampling algorithms, such as Tree-structured Parzen Estimator (TPE), to efficiently explore the hyperparameter space. Unlike grid search, which exhaustively tests predefined combinations, Optuna dynamically selects promising regions based on past evaluation results, leading to faster convergence [47]. It also implements pruning strategies, such as MedianPruner, to stop unpromising trials early, reducing computation time. In this study, Optuna was used to optimize key hyperparameters for LightGBM, XGBoost, and Random Forest models, improving performance while minimizing overfitting.

**2.6.6 Model predictions uncertainty quantification Bootstrapping.** Bootstrapping [48,49] is a resampling technique used to quantify the uncertainty of predictions made by regression models, including those built with algorithms like LightGBM. The process involves generating multiple datasets by sampling with replacement from the original training data. For each resampled dataset, a separate LightGBM model is trained. When making predictions, the ensemble of these models provides a distribution of predicted values for each input. By analyzing this distribution—specifically, by computing percentiles such as the 2.5th and 97.5th percentiles—one can construct prediction intervals that reflect the uncertainty associated with the model's predictions. This approach captures variability due to both the data and the model, offering a more comprehensive understanding of prediction confidence.

**Table 7. Metrics for Regression evaluation.**

| Metric | Definition | Meaning |
|---|---|---|
| $R^2$ | $R^2 = 1 - \frac{\sum_1^n (y_i - \hat{y}_i)}{\sum_1^n (y_i - \bar{y})}$ | $R^2 = 1$ : Model explains all variability of target variable<br>$R^2 = 0$ : Model explains none variability of target variable<br>$R^2 < 0$ : Model performs worse than the mean |
| RMSE | $RMSE = \sqrt{\frac{\sum_{i=1}^n (y_i - \hat{y}_i)^2}{n}}$ | $y_i$ : Actual values of target variable<br>$\hat{y}_i$ : Predicted values of target variable<br>$n$ : number of samples of the variable |
| MAE | $MAE = \frac{1}{n} \sum_1^n |y_i - \bar{y}|$ | The lower, the better |
| MAPE (%) | $MAPE(\%) = \frac{MAE}{\bar{y}} \cdot 100$ | The lower, the better |
| Accuracy (%) | $Accuracy = 100 - MAPE$ | The higher, the better |

**2.6.7 SHAP values and variable permutation tests.** SHAP values are an interpretability technique that explains how each feature contributes to a machine learning model's prediction. Based on cooperative game theory developed by Lloyd Shapley, they assign an important value to each feature, quantifying its contribution to the prediction.

Permutation variable tests in machine learning involve shuffling the values of a feature (or the target labels) and measuring how much the model's performance degrades. If permuting a feature leads to a significant drop in performance, that feature is considered important for the model predictions. This approach is model-agnostic and can be used for both interpreting feature importance and testing the statistical significance.

## 3. Results and discussion

### 3.1 Model evaluation and selection

Several algorithms for regression were tried training individual models. LightGBM, XGBoost, Adaboost and Random Forest Regressor (RFR).

SVR (Support Vector Machine for regression) and GradientBoost models, the latter being quite similar to XGBoost and implemented in scikit-learn were also tested but because very poor results in the first case and very similar to XGBoost result in the second case they are not included here. In the following table (Table 8) we present the results for the "basic" (without Hyperparameter optimization) models for each target variable.

LightGBM remains the best-performing model across the tests except in the case of UFL/ha model that RFR obtain a slightly better result in $R^2$ and RMSE. LightGBM and XGBoost will be optimized in their parameters in the following tests.

**Table 8. Metrics obtained for different regression algorithms and target variables.**

| Target: Dry Matter yield (kg DM/ha) | | | |
|---|---|---|---|
| **LightGBM** | **XGBoost** | **AdaBoost** | **RFR** |
| $R^2$: 0.859 | $R^2$: 0.833 | $R^2$: 0.762 | $R^2$: 0.850 |
| RMSE: 1232.4 | RMSE: 1340.1 | RMSE: 1603.3 | RMSE: 1270.5 |
| MAE: 924.8 | MAE: 1008.34 | MAE: 1217.2 | MAE: 968.6 |
| MAPE(%): 5.34 | MAPE(%): 5.78 | MAPE(%): 7.14 | MAPE(%): 5.54 |
| Accuracy(%): 94.66 | Accuracy(%): 94.22 | Accuracy(%): 92.86 | Accuracy(%): 94.46 |
| **Target: UFL yield (UFL/ha)** | | | |
| **LightGBM** | **XGBoost** | **AdaBoost** | **RFR** |
| $R^2$: 0.835 | $R^2$: 0.823 | $R^2$: 0.731 | $R^2$: 0.837 |
| RMSE: 1204.0 | RMSE: 1244.5 | RMSE: 1535.4 | RMSE: 1194.0 |
| MAE: 910.5 | MAE: 923.8 | MAE: 1186.1 | MAE: 899.5 |
| MAPE(%): 5.97 | MAPE(%): 6.03 | MAPE(%): 8.00 | MAPE(%): 5.90 |
| Accuracy(%): 94.03 | Accuracy(%): 93.97 | Accuracy(%): 92.00 | Accuracy(%): 94.10 |
| **Target: Crude Protein yield (kg CP/ha)** | | | |
| **LightGBM** | **XGBoost** | **AdaBoost** | **RFR** |
| $R^2$: 0.851 | $R^2$: 0.828 | $R^2$: 0.694 | $R^2$: 0.829 |
| RMSE: 91.7 | RMSE: 98.6 | RMSE: 131.5 | RMSE: 98.3 |
| MAE: 66.8 | MAE: 1.9 | MAE: 103.5 | MAE: 72.6 |
| MAPE(%): 6.96 | MAPE(%): 7.39 | MAPE(%): 10.37 | MAPE(%): 7.57 |
| Accuracy(%): 93.04 | Accuracy(%): 92.61 | Accuracy(%): 89.63 | Accuracy(%): 92.43 |

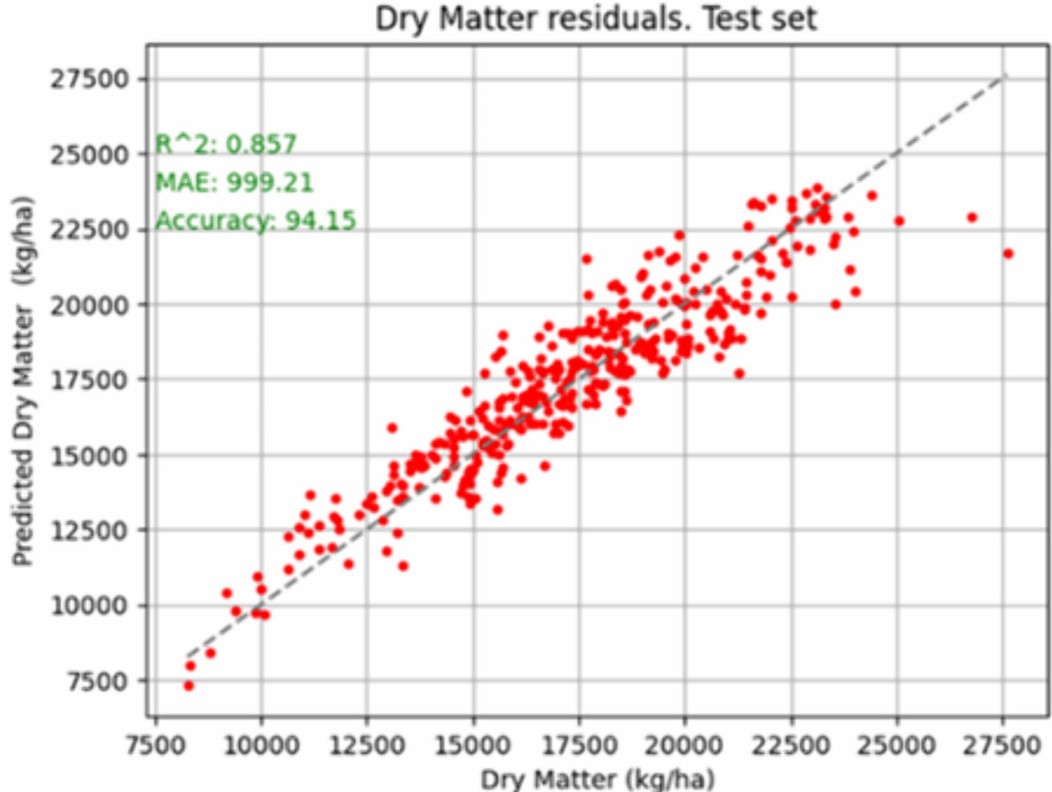

## 3.2 Residual analysis and variable importance during training

The visual representation of the differences between the actual values and the predicted values by the model in Fig 5 also allows us to visually assess the expected quality of the predictions. A perfect prediction would imply that all the red points in the graphs of Fig 5 would be situated on the diagonal line of zero difference.

In Fig 5, we can see that, in general, the prediction for Dry Matter yield model aligns quite well with the dashed line (the zero residual line). The other models, kg CP/ha and UFL/ha, (graphics not shown here for brevity), show that the predictions are also quite accurate, with small errors in the predicted values, although there is also some scatter. For all the target variables, the actual and predicted values were consistent, and the dispersion in the graph increased with the magnitudes of the values to be predicted. However, as can be seen in Table 8, the relative MAPE values in percentage are very similar, and similar relative errors in predictions are expected for all three models.

In Fig 6, we observe that the variables with the most significant influence on the training of models for all three target variables, consistently the Growing season and Radiation. Solar radiation constitutes the primary energy source for crop production. Cloudy, rainy periods that limit the amount of solar radiation available to a maize crop during susceptible stages of development contribute to regional differences that can have significant effects on yield [50]). Crop productivity (kg DM/ha) is a function of the amount of the Photosynthetically Active Radiation (PAR) absorbed or intercepted by the crop, which depends on incident PAR radiation and radiation use efficiency (RUE, units of g/MJ PAR) in the period of time in which it is grown, assuming that other factors are not limiting or conditioning (pests, diseases, water, nutrients, etc.) [51].

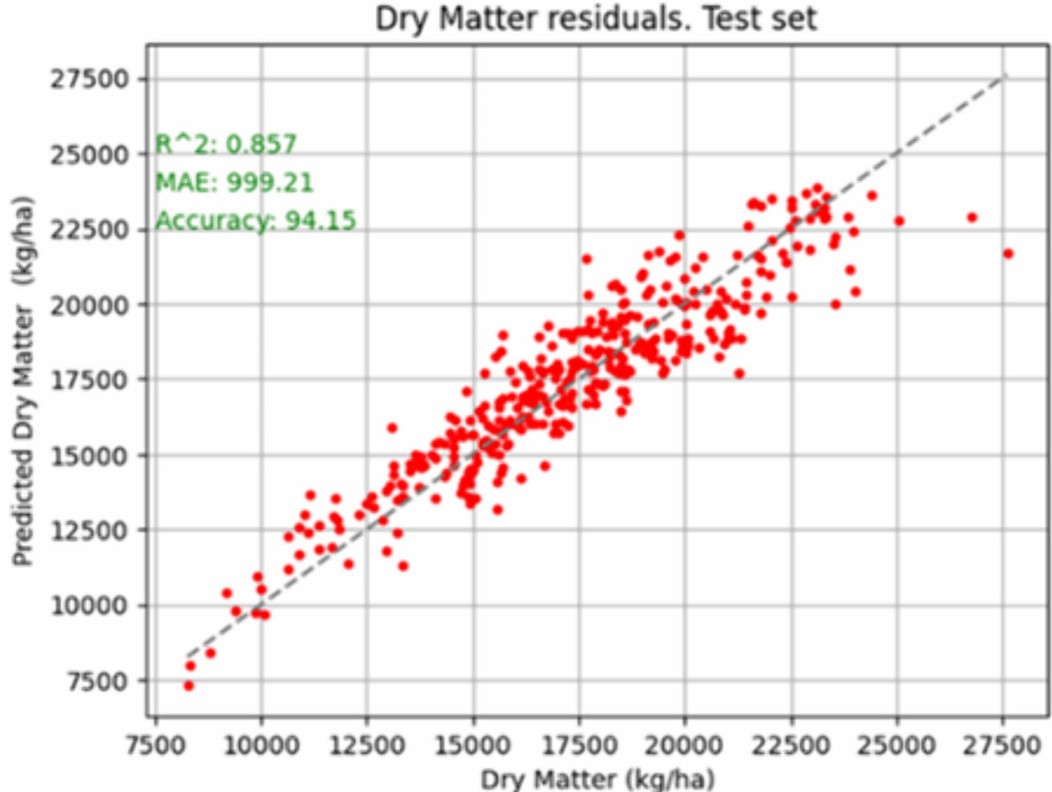

**Fig 5. Scatter plot of observed vs. predicted target Dry Matter yields (kg DM/ha).** Solid line represents the 1:1 relationship between observed and predicted yields. Model comparison metrics are $R^2$, Accuracy and Mean Absolute Error (MAE).

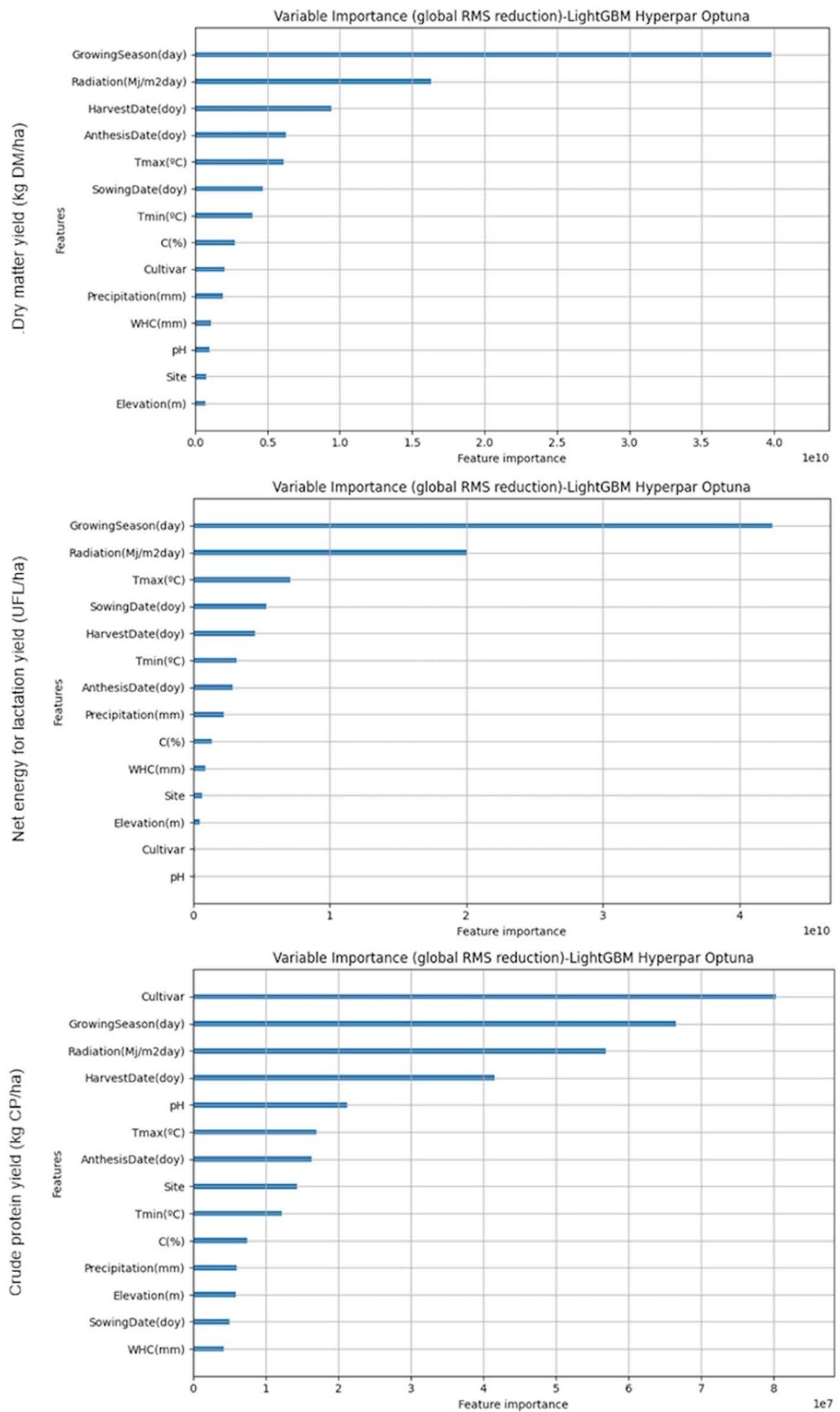

**Fig 6. Relative importance of the variables, calculated for LightGBM Optimized models of dry matter yield (kg DM/ha), net energy for lactation yield (UFL/ha) and crude protein yield (kg CP/ha).**

Beyond these, the influence of other variables varies depending on the specific target variable. However, their impact on the overall reduction of Root Mean Square Error (RMSE) is comparatively minor. This indicates that while secondary variables contribute to the model performance, their effect on improving prediction accuracy is limited compared to the primary factors.

Effects of temperature on biomass production and its components, radiation interception and RUE are twofold. First, and most important, temperature changes the duration of the period from sowing to harvest (Growing season). At high latitudes in Europe, Asia and North America, warming over recent decades has extended this period, with positive implications for crop growth and yield. Second, RUE is non-linearly related to temperature, an effect that is mediated by the effects of temperature on leaf gross photosynthesis, respiration and dry matter partitioning [52].

Maximum temperatures, which appear in fifth or sixth position as influential variable, were always below 30 ºC in our study. Maize plants are sensitive to heat stress (>30 ºC) and there is a strong decline in grain yield above this temperature when maintained for a long time [53].

The growing cycle of the FAO maize cultivars (200, 300 and 400) depends on the thermal time, i.e., the sum of temperatures that the maize accumulates each day from the day of sowing to the day of harvest (maize silage) or until the day of physiological maturity (maize grain).

Each maize cultivar has its own thermal time; the number of days needed to reach this thermal time (related to the Growing Season) varies every year. Several studies have quantified the impact of climate change, in particular the increase in temperatures on maize cultivation in Spain [54,55]. The findings of these studies show that the increase in temperature causes a decrease in yield, even under non-water-limiting conditions, due to the shortened growing cycle. Thus, for maize forage and a given sowing date and site in a warmer than normal summer, the time to harvest will be shorter (fewer days), while in a summer with cooler than normal temperatures, the time to harvest will be longer (more days). Therefore, for the FAO cultivars (200, 300 and 400) and a given sowing date and site, a long growing cycle will be more advantageous in hot summers, and a short growing cycle will be more advantageous in cooler summers.

## 3.3 Hyperparameter optimization

The LightGBM method provides the best results, along with XGBoost. These two models, together with the Random Forest Regressor, have undergone hyperparameter optimization, yielding the following results for kg DM/ha, UFL/ha and kg CP/ha predictions.

The improvements achieved through hyperparameter optimization are modest in terms of both R² and RMSE, but LightGBM remains the best-performing model.

The two variables that strongly influence predictions across all cases are Growing season and Radiation. These analyses were performed on the specified models after their hyperparameters had been optimized using the Optuna package.

Hyperparameter tuning did not yield a significant improvement over the base models (Table 9). The model for kg CP/ha has a lower absolute MAE and is expected to make better predictions for this target variable than UFL/ha and kg DM/ha, even though the R² score for kg CP/ha is slightly lower. However, the three models are almost equivalent considering that the kg CP/ha variable is one order of magnitude lower than the other two. Nonetheless, all three models explained more than 86% of the variability in the data, with a relative error of about 6-7%, (with respect to the mean of the target variable). which we believe is a remarkable result.

## 3.4 Variable permutation tests

To assess the influence of the predictor variables on the model predictions variable permutation test were performed (Fig 7).

The results of the permutation tests confirm that the biggest influence variables on predictions are Growing season and Radiation except for kg CP/ha, because Crude protein is very influenced by the Harvest date.

**Table 9. Metrics obtained for different regression algorithms after Hyperparamter Optimization.**

**kg DM/ha models**

| LightGBM | XGBoost | RFR |
|---|---|---|
| R²: 0.863 | R²: 0.862 | R²: 0.848 |
| RMSE: 1212.4 | RMSE: 1219.4 | RMSE: 1278.8 |
| MAE: 914.7 | MAE: 911.33 | MAE: 1004.33 |
| MAPE(%): 5.25 | MAPE(%): 5.27 | MAPE(%): 5.83 |
| Accuracy(%): 94.75 | Accuracy(%): 94.73 | Accuracy(%): 94.17 |

**UFL/ha models**

| LightGBM | XGBoost | RFR |
|---|---|---|
| R²: 0.840 | R²: 0.841 | R²: 0.832 |
| RMSE: 1186.0 | RMSE: 1182.7 | RMSE: 1214.4 |
| MAE: 886.2 | MAE: 889.9 | MAE: 928.1 |
| MAPE(%): 5.77 | MAPE(%): 5.80 | MAPE(%): 6.13 |
| Accuracy(%): 94.13 | Accuracy(%): 94.20 | Accuracy(%): 93.87 |

**kg CP/ha models**

| LightGBM | XGBoost | RFR |
|---|---|---|
| R²: 0.855 | R²: 0.852 | R²: 0.845 |
| RMSE: 90.5 | RMSE: 91.4 | RMSE: 93.7 |
| MAE: 65.9 | MAE: 66.4 | MAE: 68.4 |
| MAPE(%): 6.85 | MAPE(%): 7.03 | MAPE(%): 7.08 |
| Accuracy(%): 93.15 | Accuracy(%): 92.97 | Accuracy(%): 92.92 |

## 3.5 SHAP values

The studied SHAP values on the trained LightGBM models explain how each feature contributes to a machine learning model's prediction (Fig 8). Based on cooperative game theory developed by Lloyd Shapley, they assign an important value to each feature, quantifying its contribution to the prediction.

For the trained LightGBM model, we can infer the following from the plot.

Growing season has a significant impact showing that high values (in red) have positive SHAP values, indicating that longer growing season contributes positively to yield. High radiation and Harvest date values (in red) also contribute positively to the prediction. Interestingly early Sowing dates (in blue) show a positive impact, whereas later dates (in red) tend to have a negative effect.

For Tmax (°C) and Tmin (°C), higher temperatures appear to have a positive impact on the prediction. Cultivar and Site categorical features show centered impact distributions without a clear directional trend.

The vertical spread of points for the same feature suggests possible interactions with other variables. For example, the vertical dispersion of SHAP values around zero for the Radiation variable, combined with mixed red and blue colors, indicates that the same SHAP value (i.e., the same impact on the prediction) can occur for different feature values (as reflected by the colors). This suggests that the effect of that variable depends on the values of other variables in the model. A similar interaction appears in Growing Season, particularly in the region where $1000 < SHAP < 2000$, possibly involving other variables as well.

## 3.6 Bootstrapping results

In Fig 9, the indices of the first 50 samples from the test set are shown on the x-axis (only the first 50 samples are plotted to improve the visibility of the lines and the confidence interval). The y-axis displays the values of dry matter yield per

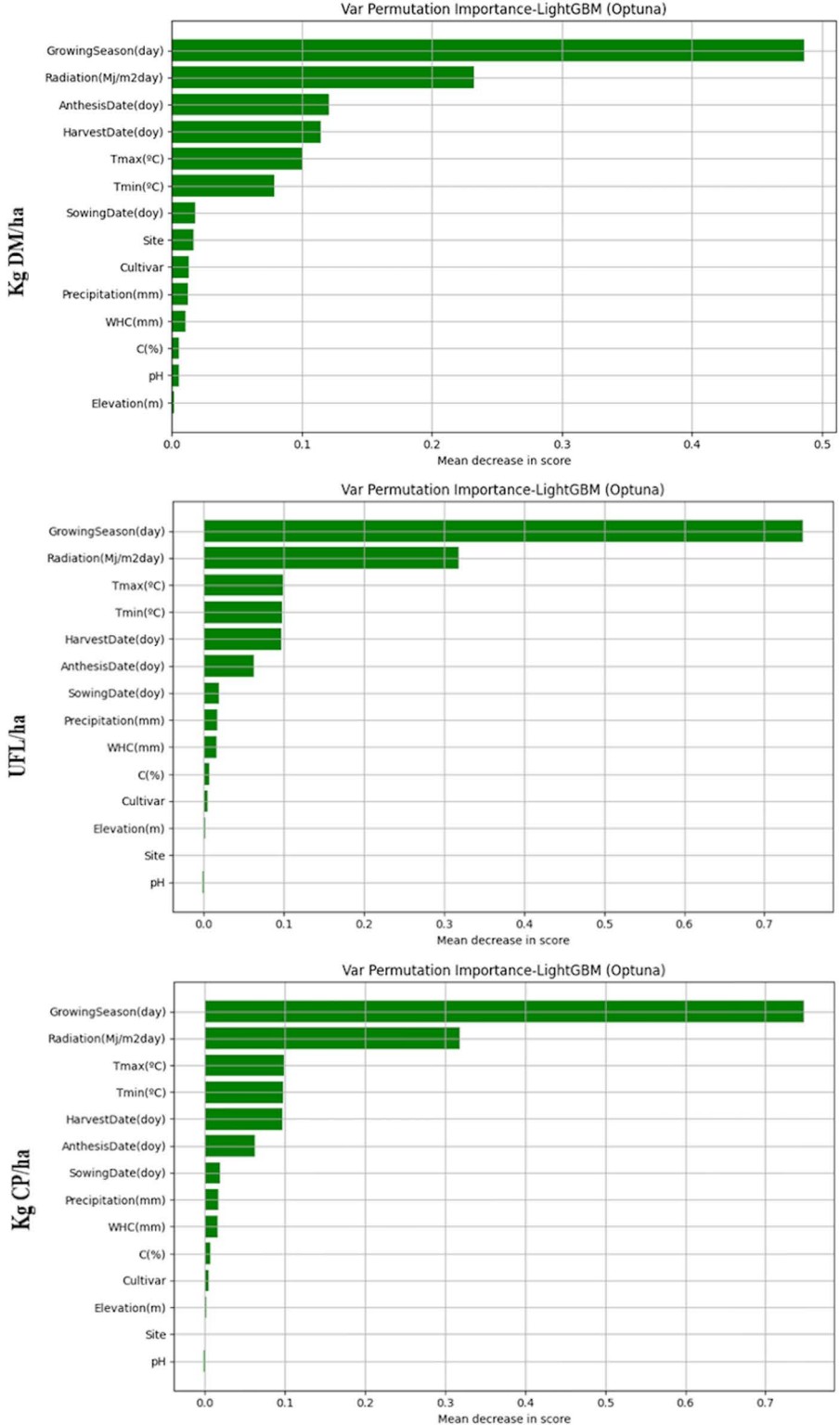

**Fig 7. Variable permutation importance test results for kg DM/ha, UFL/ha and kg CP/ha predictions.**

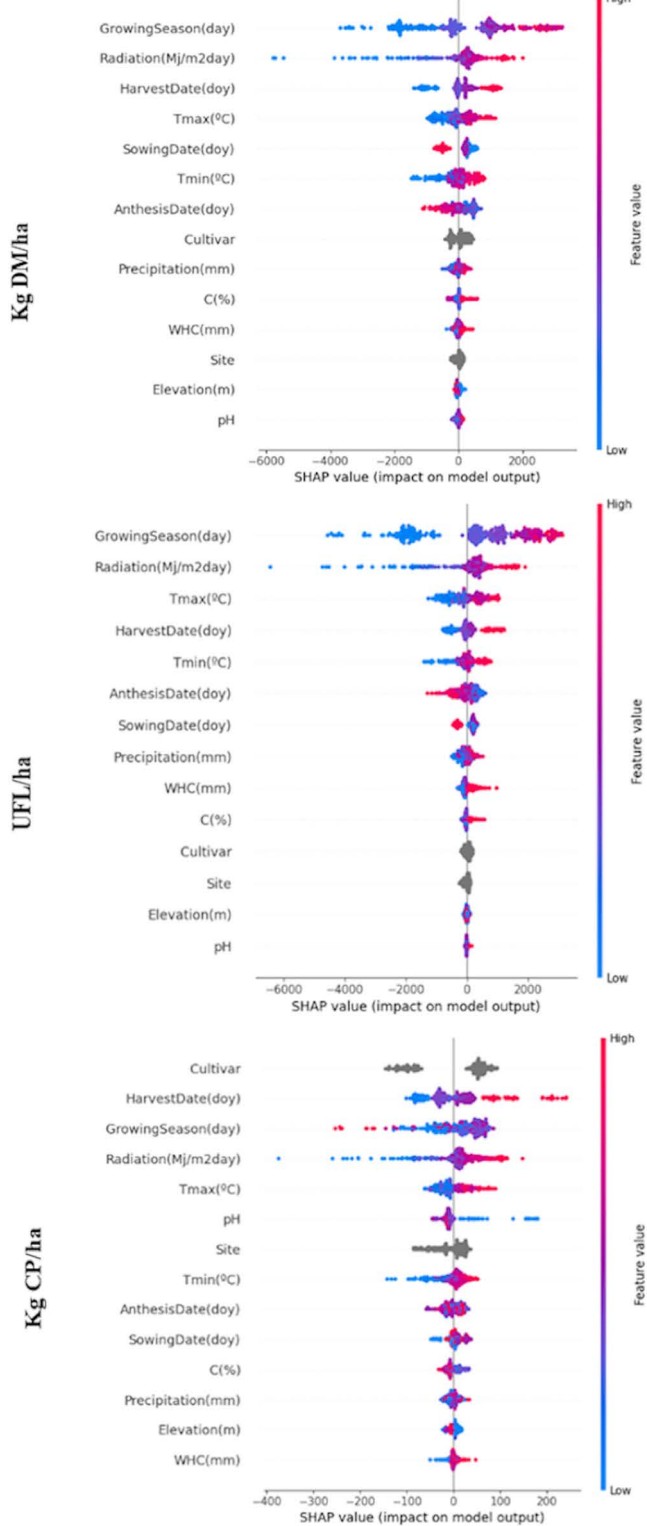

**Fig 8. SHAP values showing the impact of every variable on predictions for kg DM/ha, UFL/ha and kg CP/ha.**

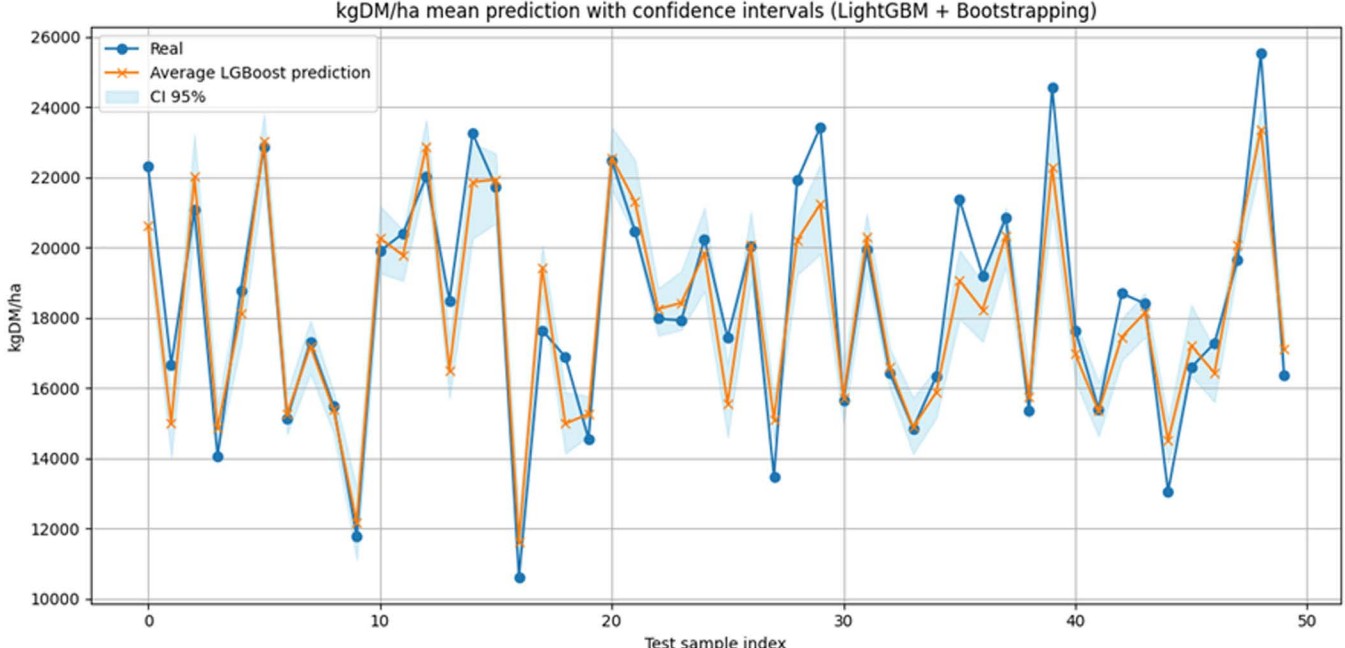

**Fig 9. Bootstrap technique results for LightGBM predictions from the test set, with 95% confidence interval band.**

hectare. The blue line represents the actual values corresponding to these samples from the test set. The orange line corresponds to the mean yield predicted by the LightGBM model during the bootstrapping process with 100 trained models. The light blue shaded band represents the 95% confidence interval (2.5% and 97.5% percentiles) associated with those mean predicted values of the orange line.

In this case, the orange line closely follows the blue one, which suggests that the model captures the general trend of the actual test data well. Furthermore, the confidence interval band is narrow, indicating that the model's confidence is high and that large variability in the predictions is not expected. The proportion of predictions for the total test set samples that fall within the 95% confidence band is approximately 55%. This relatively low percentage may suggest that there is still some bias in the data that the model has not been able to capture, or it could correspond to noise in the original data. This also corresponds to the value of the variance not modeled by the LightGBM training, as indicated by the $R^2$ metric of 0.86.

### 3.7 Prediction web application

To make the predictive capabilities of the developed models available to the public, a web application has been created in which users can obtain predictions of kg DM/ha, UFL/ha, and kg CP/ha by first choosing some data.

In Fig 10 the main interface of this app can be seen.

### 3.8 Influence of Growing season and Radiation on Dry matter yield

Growing season and Radiation were the variables that most influence both the global reduction of the Root mean square error (RMSE) and the predictions (as obtained in the new variable permutation tests performed. We also calculated a set of predictions (30 by each Site studied) under conditions considered favorable for these two variables (i.e., Growing season and Radiation above their respective mean values), unfavorable (i.e., Growing season and Radiation below their means), and intermediate conditions when neither of the two thresholds—high or low—are simultaneously met. The

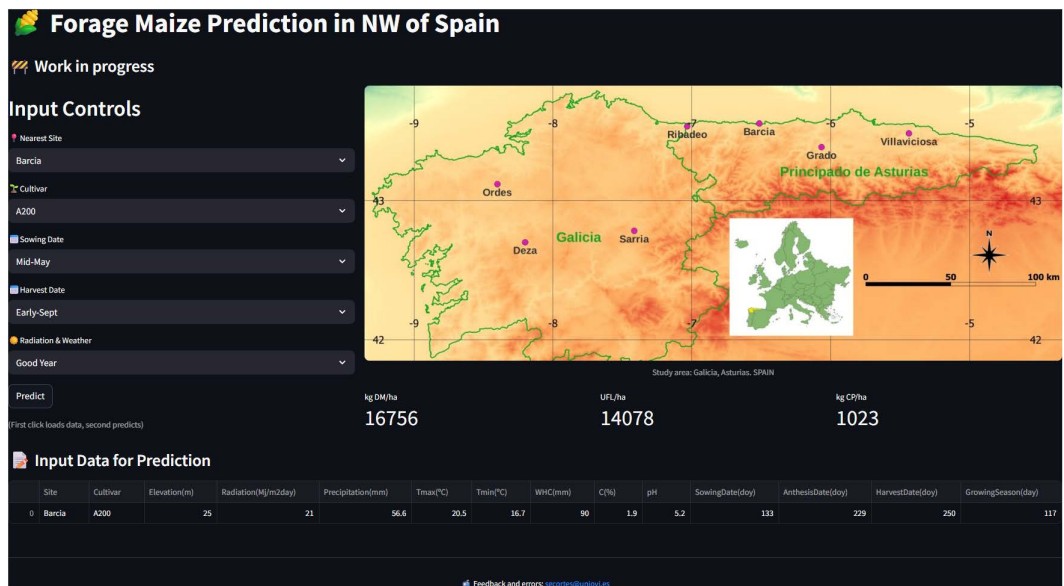

**Fig 10. Prediction web app.** (Link for this app is supplied in the Supporting information section in S1 File).

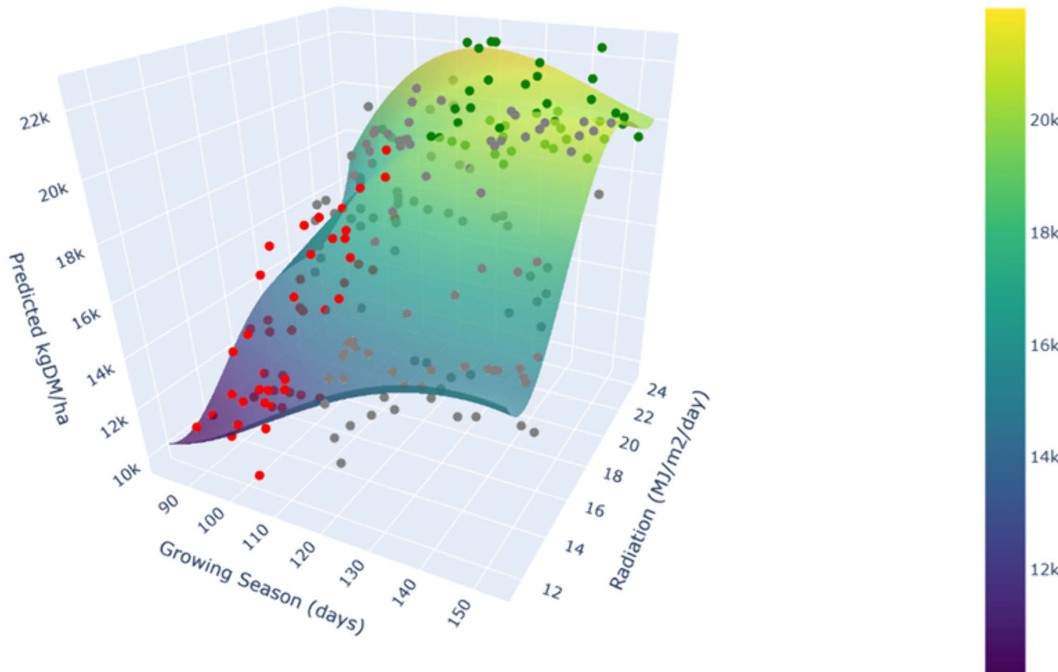

**Fig 11. Effect of Growing Season (GS) and Radiation (Rad) on predicted dry matter yield (kg DM/ha).** Horizontal axes cover the full range of GS and Rad values used to train the model.

remaining predictor variables were assigned their overall mean values, and for the categorical variable "Site", 30 samples were taken for each of the seven locations with different "Cultivar" values. The result can be seen in Fig 11.

To better appreciate the result, we have developed a web app (the link for that tool is supplied in Supporting information section in S1 File) where the user can rotate and interact with the 3D Fig 11, as well as show or hide the interpolating polynomial surface or change its degree, check numerical data.

The model shows a robust response, yielding higher outputs under favorable conditions for these two variables, and the opposite behavior under unfavorable conditions. This indicates stability in the predictions, with no signs of excessive variability or erratic outcomes.

## 4. Conclusions

The Machine Learning models obtained have the potential use as a tool to evaluate production and quality of forage maize as affected by the geographical location of the field, cultivar type, sowing and harvesting dates, and assumptions regarding weather variables during the growing season.

This paper presents a procedure that allows the creation of these predictive models of yield and nutritional quality, which could be applied to other types of crops and locations.

To our knowledge, there are currently no other models for forage maize available for this geographic area. It is important to note that for cultivars other than those used in this study, it will be necessary to test and train a new Machine learning model based on the new data available.

These three prediction models (one LightGBM model for each target variable) are implemented on a public web app tool. The link for that tool is supplied in the Supporting information section in S1 File.

## Supporting information

**S1 File.**
(DOCX)

## Acknowledgments

The authors are grateful for a grant provided by the "OECD Co-operative Research Programme" to support a research visit by the second author at the University of Florida, Gainesville, Florida, USA in 2022. Also gratefully acknowledge the reviewers for their insightful comments and constructive suggestions, which have greatly improved the quality of this manuscript. Finally, we acknowledge with gratitude the assistance of Ms. Christine Francis in the preparation of this paper in English.

## Author contributions

**Conceptualization:** Silverio Garcia-Cortes, Agustín Menéndez-Díaz, José Alberto Oliveira-Prendes.

**Data curation:** Silverio Garcia-Cortes, Agustín Menéndez-Díaz, María José Bande-Castro, Alfonso Carballal-Samalea, Adela Martínez-Fernández, José Alberto Oliveira-Prendes.

**Formal analysis:** Silverio Garcia-Cortes, Agustín Menéndez-Díaz, María José Bande-Castro, Alfonso Carballal-Samalea, Adela Martínez-Fernández, José Alberto Oliveira-Prendes.

**Funding acquisition:** Silverio Garcia-Cortes, María José Bande-Castro, Alfonso Carballal-Samalea, Adela Martínez-Fernández, José Alberto Oliveira-Prendes.

**Investigation:** Silverio Garcia-Cortes, Agustín Menéndez-Díaz, José Alberto Oliveira-Prendes.

**Methodology:** Silverio Garcia-Cortes, Agustín Menéndez-Díaz, José Alberto Oliveira-Prendes.

**Project administration:** Silverio Garcia-Cortes, José Alberto Oliveira-Prendes.

**Resources:** Silverio Garcia-Cortes, Agustín Menéndez-Díaz, María José Bande-Castro, Alfonso Carballal-Samalea, Adela Martínez-Fernández, José Alberto Oliveira-Prendes.

**Software:** Silverio Garcia-Cortes, Agustín Menéndez-Díaz, José Alberto Oliveira-Prendes.

**Supervision:** Silverio Garcia-Cortes, Agustín Menéndez-Díaz, José Alberto Oliveira-Prendes.

**Validation:** Silverio Garcia-Cortes, Agustín Menéndez-Díaz, José Alberto Oliveira-Prendes.

**Visualization:** Silverio Garcia-Cortes, Agustín Menéndez-Díaz, José Alberto Oliveira-Prendes.

**Writing – original draft:** Silverio Garcia-Cortes, Agustín Menéndez-Díaz, José Alberto Oliveira-Prendes.

**Writing – review & editing:** Silverio Garcia-Cortes, José Alberto Oliveira-Prendes.

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
