## [Decision Letter · Decision Letter 0]

5 Dec 2024

Dear Dr. Garcia-Cortes,

Authors should make adjustments suggested by the Reviewer.

We look forward to receiving your revised manuscript.

Kind regards,

Adalberto Benavides-Mendoza, Ph.D.

Academic Editor

PLOS ONE

“This research was supported by a mobility research grant awarded to JAO (TAD/CRP PO 500109615) from the OECD Co-operative Research Programme. The funders had no role in study design, data collection and analysis, decision to publish, or preparation of the manuscript.”

5. Thank you for stating the following in the Funding Section of your manuscript:

“This research was supported by a grant (TAD/CRP PO 500109615) from the OECD Co-operative Research Programme The funders had no role in study design, data collection and analysis, decision to publish, or preparation of the manuscript.”

“This research was supported by a mobility research grant awarded to JAO (TAD/CRP PO 500109615) from the OECD Co-operative Research Programme. The funders had no role in study design, data collection and analysis, decision to publish, or preparation of the manuscript.”

Additional Editor Comments:

Authors should make adjustments suggested by the Reviewer.

Reviewers' comments:

Reviewer's Responses to Questions

**Comments to the Author**

1. Is the manuscript technically sound, and do the data support the conclusions?

Reviewer #1: Yes

2. Has the statistical analysis been performed appropriately and rigorously?

Reviewer #1: Yes

3. Have the authors made all data underlying the findings in their manuscript fully available?

Reviewer #1: No

4. Is the manuscript presented in an intelligible fashion and written in standard English?

Reviewer #1: Yes

Reviewer #1: Manuscript ID: PONE-D-24-46681

Title: A machine learning approach for estimating forage maize yield and quality in NW Spain

The subject is relevant and interesting, while it needs major improvements. Overall, I can say that the manuscript has been written well, while its presentation is not fine in my opinion and can be significantly improved.

COMMENTS

- Page 3, L41: It is surely needed adding a new table to present calibrated genetic coefficients for each hybrid, in which, if you used calibrated genetic coefficients from other studies, they have to be presented properly.

- Page 6, L3: I did not find any formulation for the Random Forest approach. All explanations are descriptive without any mathematical presentation.

- The captions for many tables and figures are not clear. For instance, in Table 2, what are theses results for? I mean for which station or site? It is very confusing. Or in Figure 8, are these results from CERES-Maize or from the RF model? This is a big problematic issue for the manuscript and must be fundamentally revised.

- In results and discussion section, add even few sentences about the gridded climate datasets and their benefits as an alternative for measured data and refer to https://doi.org/10.1007/s12145-023-01215-0. It is important because some readers might be interested to conduct similar study as yours in their region but they cannot because of absence of measure weather data.

- I encourage the authors to spend more time for the preparation of the figures, because they can be much better. They are very simple.

- Adding flowchart for the study workflow can improve the readability of the methodology.

**Do you want your identity to be public for this peer review?** For information about this choice, including consent withdrawal, please see our Privacy Policy

Reviewer #1: No

---

## [Author Response · Author response to Decision Letter 1]

30 Jan 2025

Response to Reviewers

Thank you very much for suggestions and corrections. We have tried to improve the document following your indications and the ones from editor. Sadly, in the main body template supplied line numbering is not working. In the following we will refer to new page numbering in the unmarked manuscript asked by the editor.

Reviewer 1. Comment responses:

-Page 3, L41: It is surely needed adding a new table to present calibrated genetic coefficients for each hybrid, in which, if you used calibrated genetic coefficients from other studies, they have to be presented properly.

Thank you for your suggestion. Now in the page 4 and 5, under “Adaptation of the CSM-CERES-Maize model” sections, two new tables have been added. Table1 explains the CSM-CERES-Maize model parameters (genetic coefficients) and the new Table 2. Shows the values for that genetic coefficients used for each studied cultivar.

Page 6, L3: I did not find any formulation for the Random Forest approach. All explanations are descriptive without any mathematical presentation.

In page 12, section Random Forest Regressor we have added a new image Fig. 7 explaining the fundamental concept of the Random Forest algorithm for regression.

Also new explanations have been added and since tree decision is the key component of a random forest, also a new Table 5 in Page 12 resume the steps for the splitting node strategy (with equations) in a tree. We copied here the main text added.

“A Decision Tree is also a supervised learning algorithm used for both classification and regression tasks. In the context of regression, it predicts continuous outcomes by recursively partitioning the data space into subsets based on feature values, aiming to minimize the variance within each subset. In regression, the objective of a decision tree is to split the data into subsets to minimize the loss at each node. The typical loss function minimized is the mean squared error (MSE). The reduction in MSE when splitting a node is known as "variance reduction." The goal is to find the split that maximizes this reduction.

where y_R1 is the mean of the R1 region and similarly y_R2. Initially, the root node receives the entire dataset of samples. As the tree grows, each split divides the data into subsets, which are then passed down to the child nodes. Splitting stops when the node contains fewer than a pre-specified minimum number of simples or when a node becomes "pure," the variance change is negligible for regression. The workflow for a decision tree can be summarized as:”

Compute the total error at the node: L_"node" =∑_(i∈R)▒(y_i-(y_R ) ® )^2

For each feature x_j and threshold t, compute: L_"split" =∑_(i∈R_1)▒(y_i-(y_(R_1 ) ) ® )^2 +∑_(i∈R_2)▒(y_i-(y_(R_2 ) ) ® )^2

Calculate the variance reduction: ΔL(j,t)=L_node-L_split for all splits and thresholds.

Choose the optimal feature j and the threshold t that maximize ΔL(j,t)

The captions for many tables and figures are not clear. For instance, in Table 2, what are theses results for? I mean for which station or site? It is very confusing. Or in Figure 8, are these results from CERES-Maize or from the RF model? This is a big problematic issue for the manuscript and must be fundamentally revised.

Thanks for your comment. New captions with (we think) better and more explanations have been added through all the document. For example, check now the captions for: Tables 1,2, (these two tables are new), and expanded captions for Table 3 and 4 and the rest of figures.

We also have clarified the structure of the manuscript. The new structure is in our opinion clearer.

Introduction

Materials y methods

Experimental sites and minimum dataset in Asturias

Experimental sites and minimum dataset in Galicia

Adaptation of CSM-CERES Maize model

Simulation of the international variation in forage maize production

Machine learning

Exploratory Analysis (EDA)

Random Forest Regressor

Results and discussion

Model fitting results

Conclusions

- In results and discussion section, add even few sentences about the gridded climate datasets and their benefits as an alternative for measured data and refer to https://doi.org/10.1007/s12145-023-01215-0. It is important because some readers might be interested to conduct similar study as yours in their region but they cannot because of absence of measure weather data.

Thanks again for this suggestion and the included link. We have a paragraph highlighting the importance of using the gridded meteorological dataset corresponding to precipitation, temperatures and solar radiation variables as reliable alternatives to directly measured data. Please check this addition in page 8.

Text added:

“ Good quality long-term station data play a significant role in characterizing the climatic

conditions and in assessing their suitability for agricultural production [29]. Recent studies highlight the importance of using gridded meteorological datasets as reliable alternatives to directly measured data, especially in areas with sparse weather station coverage [30]. Examples of available meteorological databases include the following: NASA Power from the NASA Langley Research Center POWER (Prediction Of Worldwide Energy Resources) Project (https://power.larc.nasa.gov/) provides solar and meteorological data sets from NASA research for support of renewable energy, building energy efficiency and agricultural needs and CHIRPS (Climate Hazards Group InfraRed Precipitation with Station data) from the Climate Hazards Center, University of California, Santa Barbara for precipitation data (https://www.chc.ucsb.edu/data/chirps) [31]

I encourage the authors to spend more time for the preparation of the figures, because they can be much better. They are very simple.

Our objective was to prepare as clear figures as possible, but we recognize that some of them were very simple figures. We have rethought the figures to achieve a more professional representation of the different variables involved. Thanks for your suggestion.

Figures numbered Fig2, Fig. 3 , Fig 4 and Fig. 5 and Fig. 6 are completely new. Fig. 2 includes a previous Exploratory analysis of the data in a visual manner and the rest till Fig. 6 are new. Fig 3 resume all the meteorological variables in only one chart.

- Adding flowchart for the study workflow can improve the readability of the methodology.

A new figure with a general flowchart has been included in the manuscript. Please check fig.1

About Code and Data availability:

Following the editor suggestion , although the data link: :https://figshare.com/articles/dataset/Forage_maize_yield_and_quality_data_in_the_NW_of_Spain_Galicia_and_Asturias_/27044752?file=49239496

included in the previous manuscript version is still working, in this second revised document we also have included a code document (as a Jupyter notebook in Python). Data file and code file are both available in this new link:

Data and code can be accessed in : https://github.com/sgcortes/ForageMaize24

---

## [Decision Letter · Decision Letter 1]

6 Apr 2025

Dear Dr. Garcia-Cortes,

Thank you for submitting your manuscript to PLOS ONE. After careful consideration, we feel that it has merit but does not fully meet PLOS ONE’s publication criteria as it currently stands. Therefore, we invite you to submit a revised version of the manuscript that addresses the points raised during the review process.

We look forward to receiving your revised manuscript.

Kind regards,

RISHIRAJ DUTTA, Ph.D.

Academic Editor

PLOS ONE

**Journal Requirements:**

**Additional Editor Comments:**

The manuscript can be considered for publication if the author(s) can address the comments from reviewers.

Reviewers' comments:

Reviewer's Responses to Questions

**Comments to the Author**

Reviewer #1: All comments have been addressed

Reviewer #2: All comments have been addressed

2. Is the manuscript technically sound, and do the data support the conclusions?

Reviewer #1: Yes

Reviewer #2: Partly

3. Has the statistical analysis been performed appropriately and rigorously?

Reviewer #1: Yes

Reviewer #2: Yes

4. Have the authors made all data underlying the findings in their manuscript fully available?

Reviewer #1: Yes

Reviewer #2: Yes

5. Is the manuscript presented in an intelligible fashion and written in standard English?

Reviewer #1: Yes

Reviewer #2: Yes

**Reviewer #1:**  The comments have been addressed and the revised manuscript looks better. So, it can be accepted now.

**Reviewer #2: ** The MS needs further revision to improve clarity, coherence, and scientific presentation.

My specific comments are:

1. In the abstract, the author mention that non-specialists can use this model. Is user interface or web-based tool developed? If so, please provide access details or describe its functionality.

2. Write keywords.

3. Introduction and Discussion sections are poorly structured and lack clarity. The authors are encouraged to refer recent paper on ensemble machine learning techniques: [https://doi.org/10.1016/j.atech.2024.100543].

4. In the Materials and Methods section, please specify the programming language, along with the libraries or packages used for data processing, modeling, and analysis.

5. Your study spans different environments and sowing dates, but no explicit interaction terms for genotype × environment × management are analyzed. Did you examine whether cultivar response varies significantly by location or sowing time in the ML outputs?

6. Did you directly use DSSAT output variables (like anthesis date, total biomass, nitrogen uptake) as predictors in the ML model, or were only weather, soil, and management variables used? Please clarify the feature engineering process.

7. Clarify the proportion of simulated vs. actual field data used for model training and testing? Have they validated the ML model using independent field-observed data?

8. Have the authors considered incorporating uncertainty quantification for model predictions? For instance, bootstrapping or Bayesian Random Forest could help assess prediction robustness.

9. Could the authors elaborate more on how cultivar and Tmax affect the predictions biologically and statistically?

10. The Random Forest algorithm are explained in too much depth for a typical agri-paper and repeated across sections. This technical detail could be moved to supplementary material to improve readability

11. Author discuss more explicitly the limitations of using only simulation-based data, and how future studies could include more real-world datasets?

12. The manuscript uses only R², MAE, and MAPE. No RMSE, no prediction intervals, no feature permutation tests, no SHAP values.

13. Why there no comparison with other ensemble machine learning techniques, such as Gradient Boosting Machines (e.g., XGBoost, LightGBM) or AdaBoost? Including such comparisons would help assess the relative performance and robustness of the Random Forest model used in this study?

14. What data splitting strategy was used to ensure no temporal data leakage?

15. Try to add characteristics of cultivars on supplementary material.

16. Share all DSSAT parameter files, calibration datasets, and pre-processed ML features (not just raw code) to allow full reproducibility of the study.

17. The Conclusion section should be more concise and focused on the practical implications of the study, particularly from an agronomic and economic perspective. Rather than emphasizing that a model was “successfully developed” the authors should highlight how the model’s predictions could support cost-effective decision-making or improve resource use efficiency in forage maize production.

**Do you want your identity to be public for this peer review?** For information about this choice, including consent withdrawal, please see our Privacy Policy

Reviewer #1: No

Reviewer #2: **Yes: ** Dr Shankar Lal Jat

---

## [Author Response · Author response to Decision Letter 2]

20 May 2025

Please check the document Answer to Reviewers for a more complete and best formatted answer (including graphics).

Reviewer #2: The MS needs further revision to improve clarity, coherence, and scientific presentation.

My specific comments are:

1. In the abstract, the author mention that non-specialists can use this model. Is user interface or web-based tool developed? If so, please provide access details or describe its functionality.

Thank you for your suggestion. This is a goal we plan to pursue once the publication of this study in a research journal such as PLOS ONE is achieved. A preliminary version of a web app tool hosted on Streamlit Cloud is already public. A capture of its main interface was included in the text on page 23. The tool is available on: https://nwspainforagemaize.streamlit.app

2. Write keywords.

Thank you for the comment. You may not have been able to access the first page containing the title and the authors (please check the PDF document from PLOS ONE). On that page, the keywords have already been provided, which are as follows:

CSM-CERES-Maize model.

decision support system.

forage yield and quality.

LightGBM (Random Forest Regressor was removed)

silage maize.

machine learning.

3. Introduction and Discussion sections are poorly structured and lack clarity. The authors are encouraged to refer recent paper on ensemble machine learning techniques: [https://doi.org/10.1016/j.atech.2024.100543].

Following your suggestion we have tried to improve the agronomic context in the Introduction section.

On page 2 the following text was included:

The CSM-CERES-Maize model [5] [6] [7] within the same DSSAT package has been widely used to support decision-making regarding the management of irrigation and fertilization, as well as the choice of maize cultivars. CERES-Maize continues to be the most widely used maize model globally and remains the basis of other maize models, including APSIM [8] and the CSM-IXIM [9].

In forage maize, [10] [11] initially adapted the CSM-CERES-Maize model (CSM = Cropping System Model) in the software provided by the Decision Support System for Agrotechnology Transfer (DSSAT) [12], in order to simulate the growth and development of three forage maize cultivars (SE1-200, SE2-300, and SE3-400) in three sites (Barcia, Villaviciosa and Grado) in Asturias.

One of the major problems in the selection of genotypes (cultivars) with high productivity in different environments (locations, years) is the genotype x environment interaction (GEI). Multi-environment trials (MET) often use the AMMI (additive main-effects and multiplicative interaction) model which is popular for analyzing MET data with fixed effect [13]. This model is a statistical tool for identifying GEI patterns and allows grouping genotypes according to response characteristics (identification of stable genotypes) and detecting trends between environments [14].

On page 18 the following text was included:

In Figure 6, we observe that the variables with the most significant influence on the training of models for all three target variables, consistently the Growing season and Radiation. Solar radiation constitutes the primary energy source for crop production. Cloudy, rainy periods that limit the amount of solar radiation available to a maize crop during susceptible stages of development contribute to regional differences that can have significant effects on yield [48]). Crop productivity (kg DM/ha) is a function of the amount of the Photosynthetically Active Radiation (PAR) absorbed or intercepted by the crop, which depends on incident PAR radiation and radiation use efficiency (RUE, units of g/MJ PAR) in the period of time in which it is grown, assuming that other factors are not limiting or conditioning (pests, diseases, water, nutrients, etc.) [49].

Beyond these, the influence of other variables varies depending on the specific target variable. However, their impact on the overall reduction of Root Mean Square Error (RMSE) is comparatively minor. This indicates that while secondary variables contribute to the model performance, their effect on improving prediction accuracy is limited compared to the primary factors.

Effects of temperature on biomass production and its components, radiation interception and RUE are twofold. First, and most important, temperature changes the duration of the period from sowing to harvest (Growing season). At high latitudes in Europe, Asia and North America, warming over recent decades has extended this period, with positive implications for crop growth and yield. Second, RUE is non-linearly related to temperature, an effect that is mediated by the effects of temperature on leaf gross photosynthesis, respiration and dry matter partitioning [50].

Maximum temperatures, which appear in fifth or sixth position as influential variable, were always below 30 ºC in our study. Maize plants are sensitive to heat stress (>30 ºC) and there is a strong decline in grain yield above this temperature when maintained for a long time [51].

The growing cycle of the FAO maize cultivars (200, 300 and 400) depends on the thermal time, i.e. the sum of temperatures that the maize accumulates each day from the day of sowing to the day of harvest (maize silage) or until the day of physiological maturity (maize grain).

Each maize cultivar has its own thermal time; the number of days needed to reach this thermal time (related to the Growing Season) varies every year. Several studies have quantified the impact of climate change, in particular the increase in temperatures on maize cultivation in Spain [52] [53]. The findings of these studies show that the increase in temperature causes a decrease in yield, even under non-water-limiting conditions, due to the shortened growing cycle. Thus, for maize forage and a given sowing date and site in a warmer than normal summer, the time to harvest will be shorter (fewer days), while in a summer with cooler than normal temperatures, the time to harvest will be longer (more days). Therefore, for the FAO cultivars (200, 300 and 400) and a given sowing date and site, a long growing cycle will be more advantageous in hot summers, and a short growing cycle will be more advantageous in cooler summers.

4. In the Materials and Methods section, please specify the programming language, along with the libraries or packages used for data processing, modeling, and analysis.

At the bottom of page 9 the following text was included:

Python has been used as the programming language within Jupyter Notebook enviromnment. The Python packages used include: numpy, pandas, for basic programming and sklearn, lightgbm, xgboost, optuna, shap for machine learning processes. Other auxiliary packages: joblib, matplotlib, streamlit, and plotly, were used for graphics and file outputs. The code is available at: https://doi.org/10.5281/zenodo.15470090

5. Your study spans different environments and sowing dates, but no explicit interaction terms for genotype × environment × management are analyzed. Did you examine whether cultivar response varies significantly by location or sowing time in the ML outputs?

Thanks for your comment. We have performed a further analysis of a three-way ANOVA to cover this remark.

On page 8 the following section was included:

2.5 Genotype × environment × management interaction

A three-way ANOVA was conducted to determine the effects of Cultivar, Site and Sowing date on dry matter yield. The three independent variables or factors (Cultivar, Site and Sowing date) were considered fixed factors.

To find out information about the three-way Site x Cultivar x Sowing date interaction (e.g., whether the three-way interaction effect is statistically significant), we need to consult the “Site x Cultivar x Sowing date" row in the Table 5.

Table 5. Summary of Three-Way Analysis of Variance for Site, Cultivar and Sowing date factors on dry matter yield (kg DM/ha)

There was no statistically significant three-way interaction between Cultivar, Site and Sowing date, F(24, 1386) = 1.287, p ≥ 0.05, but all the two-way interactions were significant.

There was a statistically significant two-way interaction effect between Cultivar and Site, on dry matter yield of the maize cultivars, F(12, 1386) = 5.288, p < 0.001. This indicates that cultivars were affected differently by the Sites. There was also a statistically significant two-way interaction effect between Site and Sowing date, F(12, 1386) = 5.775, p < 0.001 and Cultivar and Sowing date, F(4, 1386) = 4.616, p < 0.001 on dry matter yield.

We may follow up and interpret the two-way interactions but not the main effects due to the statistical significance of the two-way interactions. Usually when we have a significant two-way interaction (e.g. Site x Cultivar), it is the effect of this interaction that is of interest, and the main effects (e.g. Site and Cultivar) are less of interest, because, in this case, we know that the effect of Cultivar changes across levels of Site. There was a statistically significant simple main effect of Site, F(6, 1386) = 201.6, p < 0.001, Cultivar, F(2, 1386) = 447.9, p < 0.001, and Sowing date, F(2, 1386) = 617.9, p < 0.001, on Dry matter yield.

The graphical analysis (not presented here) showed non-crossover interactions [27] indicating that the difference in performance (kg DM/ha) of the Cultivars is not similar across the other factors (Sites or Sowing dates) but it does not change the order (ranking) of the one that produces more and the one that produces less according to the Sites or Sowing dates.

More info is reflected here. We have not included these figures in the text but are presented here to complete the study.

Graphical analysis of double interactions not included in the text, is presented below:

We may follow up and interpret the two-way interactions but not the main effects due to the statistical significance of the two-way interactions. Usually when we have a significant two-way interaction (e.g. Site x Cultivar), it is the effect of this interaction that is of interest, and the main effects (e.g. Site and Cultivar) are less of interest, because, in this case, we know that the effect of Cultivar changes across levels of Site.

There was a statistically significant simple main effect of Site, F(6, 1386) = 201.6, p < 0.001, Cultivar, F(2, 1386) = 447.9, p < 0.001, and Sowing date, F(2, 1386) = 617.9, p < 0.001, on Dry matter yield.

We plot the Means of Dry matter yield (kg DM/ha) as a function of the interactions of Cultivar and Site. We repeat the procedure for the interaction Site and Sowing date, and the interaction Cultivar and Sowing date.

The three figures showed non-crossover interactions (Allard and Bradshaw, 1964) indicating that the difference in performance (kg DM/ha) of the Cultivars is not similar across the other factors (Sites or Sowing dates) but it does not change the order (ranking) of the one that produces more and the one that produces less according to the Sites or Sowing dates.

In Figure ?, the latest cultivar 400 produces more than the other cultivars in the seven sites with different yields according to the site, with the Villaviciosa site having higher yields.

In Figure ??, the earliest sowing date 133 (in days of the year) is the most productive across all sites with different yields according to the site,

In Figure ???, the latest cultivar 400 produces more than the other cultivars on the three sowing dates with different yields according to the sowing date.

6. Did you directly use DSSAT output variables (like anthesis date, total biomass, nitrogen uptake) as predictors in the ML model, or were only weather, soil, and management variables used? Please clarify the feature engineering process.

Only the variables in the original excel file were used. These variables were: ['Site', 'Cultivar', 'Elevation (m)', 'Radiation (Mj/m2day)', 'Precipitation (mm)', 'Tmax (ºC)', 'Tmin (ºC)', 'WHC (mm)', 'C (%)', 'pH',' Sowing dates (Julian days)', 'Anthesis dates (Julian days)', 'Harvest date (Julian days)', 'Growing season (days)']. A detailed description of these variables included in the Asturian and Galician forage maize dataset generated with the CSM-CERES-Maize model were presented in Table 4.

Oliveira et al. (2023) initially adapted the CSM-CERES-Maize model in the software provided by the Decision Support System for Agrotechnology Transfer (DSSAT) for forage maize, to simulate the growth and development of three forage maize cultivars (SE1-200, SE2-300, and SE3-400) in three sites (Barcia, Villaviciosa and Grado) in Asturias. Moreover, DSSAT-based seasonal analysis was conducted to examine the interannual variation in forage maize productivity in combination with meteorological data available for 23 years (2000-2022) in the three sites in Asturias. Similar work was later performed with data from three cultivars (XU1-200, XU2-300, and XU3-400) in four sites (Ribadeo, Ordes, Deza and Sarria) in Galicia (Oliveira and Bande, 2024). The workflow carried out is summarized in Figure 1 of the manuscript.

7. Clarify the proportion of simulated vs. actual field data used for model training and testing?

The field data were observed during the years 2012–2013–2014 in Asturias and 2014–2015–2016 in Galicia. Out of the total 23 years of data, 6 years correspond to field observations and the rest are simulated, meaning that approximately 25% of the data are field based. This proportion of field data versus simulated data is expected to be maintained in both the training and test sets.

The above explanation was included now in Section 2.6 Machine Learning Processing (page 9)

Have they validated the ML model using independent field-observed data?

Sadly, no additional field data for the cultivars studied are available for years outside the study period (2012-2016). New cultivars should be then considered in future studies.

8. Have the authors considered incorporating uncertainty quantification for model predictions? For instance, bootstrapping or Bayesian Random Forest could help assess prediction robustness.

Following your suggestion, we have implemented the Bootstrap technique with the LightGBM model, which achieved the best results. Thank you for this suggestion, which we greatly appreciate. A new section 3.6 (page 23) has now been added about the results of this technique.

Also, the following text has been added to the manuscript (page 23)

In Figure 9, the indices of the first 50 samples from the test set are shown on the x-axis (only the first 50 samples are plotted to improve the visibility of the lines and the confidence interval). The y-axis displays the values of dry matter yield per hectare. The blue line represents the actual values corresponding to these samples from the test set. The orange line corresponds to the mean yield predicted by the LightGBM model during the bootstrapping process with 100 trained models. The light blue shaded band represents the 95% confidence interval (2.5% and 97.5% percentiles) associated with those mean predicted values of the orange line.

In this case, the orange line closely follows the blue one, which suggests that the model captures the general trend of the actual test data well. Furthermore, the confidence interval band is narrow, indicating that the model’s confidence is high and that large variability in the predictions is not expected. The proportion of predictions for the total test set samples that fall within the 95% confidence band is approximately 55%. This relatively low percentage may suggest that there is still some bias in the data that the model has not been able to capture, or it could correspond to noise in the original data. This also corresponds to the value of the variance not modeled by the LightGBM training, as indicated by the R² metric of 0.86.

9. Could the authors elaborate more on how cultivar and Tmax affect the predictions biologically and statistically?

After the work carried out based on the suggestions received during this review, another small app (to allow interactive exploration) for a basic sensitivity analysis of Dry matter prediction based on the values of Growing season and Radiation were created.

We believe it can give an approximate idea of what the upcoming web app

---

## [Editor Report · Decision Letter 2]

29 May 2025

A machine learning approach for estimating forage maize yield and quality in NW Spain

PONE-D-24-46681R2

Dear Dr. Garcia-Cortes,

We’re pleased to inform you that your manuscript has been judged scientifically suitable for publication and will be formally accepted for publication once it meets all outstanding technical requirements.

Kind regards,

RISHIRAJ DUTTA, Ph.D.

Academic Editor

PLOS ONE
---

## [Editor Report · Acceptance letter]

PONE-D-24-46681R2

PLOS ONE

Dear Dr. Garcia-Cortes,

I'm pleased to inform you that your manuscript has been deemed suitable for publication in PLOS ONE. Congratulations! Your manuscript is now being handed over to our production team.

Kind regards,

on behalf of

Dr. RISHIRAJ DUTTA

Academic Editor

PLOS ONE